# Skeletal muscle PGC-1α1 reroutes kynurenine metabolism to increase energy efficiency and fatigue-resistance

Leandro Z. Agudelo[1,5,6], Duarte M.S. Ferreira [1,6], Shamim Dadvar[1,6], Igor Cervenka[1], Lars Ketscher [1], Manizheh Izadi[1], Liu Zhengye[2], Regula Furrer[3], Christoph Handschin [3], Tomas Venckunas[4], Marius Brazaitis[4], Sigitas Kamandulis[4], Johanna T. Lanner [2] & Jorge L. Ruas [1]

The coactivator PGC-1α1 is activated by exercise training in skeletal muscle and promotes fatigue-resistance. In exercised muscle, PGC-1α1 enhances the expression of kynurenine aminotransferases (Kats), which convert kynurenine into kynurenic acid. This reduces kynurenine-associated neurotoxicity and generates glutamate as a byproduct. Here, we show that PGC-1α1 elevates aspartate and glutamate levels and increases the expression of glycolysis and malate-aspartate shuttle (MAS) genes. These interconnected processes improve energy utilization and transfer fuel-derived electrons to mitochondrial respiration. This PGC-1α1-dependent mechanism allows trained muscle to use kynurenine metabolism to increase the bioenergetic efficiency of glucose oxidation. Kat inhibition with carbidopa impairs aspartate biosynthesis, mitochondrial respiration, and reduces exercise performance and muscle force in mice. Our findings show that PGC-1α1 activates the MAS in skeletal muscle, supported by kynurenine catabolism, as part of the adaptations to endurance exercise. This crosstalk between kynurenine metabolism and the MAS may have important physiological and clinical implications.

---

[1] Department of Physiology and Pharmacology, Molecular and Cellular Exercise Physiology, Karolinska Institutet, Biomedicum C5, 171 77 Stockholm, Sweden. [2] Department of Physiology and Pharmacology, Molecular Muscle Physiology and Pathophysiology, Karolinska Institutet, Biomedicum C5, 171 77 Stockholm, Sweden. [3] Biozentrum, University of Basel, Klingelbergstrasse 50/70, CH-4056 Basel, Switzerland. [4] Institute of Sports Science and Innovations, Lithuanian Sports University, Sporto str. 6, 44221 Kaunas, Lithuania. [5] Present address: Computer Science and Artificial Intelligence Laboratory, Massachusetts Institute of Technology, Cambridge, MA 02139, USA. [6] These authors contributed equally: Leandro Z. Agudelo, Duarte M.S. Ferreira, Shamim Dadvar. Correspondence and requests for materials should be addressed to J.L.R. (email: jorge.ruas@ki.se)

The *Pgc-1α* gene encodes several transcriptional coactivator proteins that coordinate the expression of gene networks involved in cellular adaptive processes[1]. Originally identified as a single transcript and protein (PGC-1α1[2]), *Pgc-1α* is now known to encode several splice isoforms mostly involved in the control of energy metabolism in several tissues[3]. One exception is the PGC-1α4 variant, which regulates skeletal muscle mass[4]. In skeletal muscle, PGC-1α1 is important for the adaptation to aerobic exercise training as it ensures that fuel supply, oxygen transport, and energy metabolism are coupled to increased exercise performance and fatigue-resistance[1]. The connection between PGC-1α1 and exercise performance has often been attributed to the increase in muscle mitochondrial biogenesis and fatty acid oxidation it mediates. This was first observed in mice with sustained PGC-1α1 expression in skeletal muscle (mck-PGC-1α1 transgenics), which show many of the adaptations to exercise without any training[5]. The elevation of *Pgc-1α1* mRNA expression in the mck-PGC-1α1 mice varies from oxidative to glycolytic muscles between 2- and 10-fold, respectively. In humans *PGC-1α* activation is achieved through endurance exercise and its transcript can be elevated within the same range[6,7].

Muscle resistance to fatigue during endurance exercise depends on glycogen storage and its mobilization to glucose oxidation, as well as on the capacity to oxidize fatty acids[8]. PGC-1α1 has been shown to promote both glycogen storage and fatty acid oxidation[9–11]. Maintaining glycolytic flux depends (among other factors) on the renewal of the cytosolic $NAD^+$ pool. This is guaranteed by the transfer of glycolysis-generated NADH reducing equivalents to lactate or into the electron transport chain (in the latter case generating ATP). Since the inner membrane of the mitochondria is not permeable to NADH, skeletal muscle uses a glycerol-3-phosphate (G3P) shuttle (G3PS) that transports electrons to Coenzyme Q, thus resulting in 1.5 ATP per NADH. Other tissues such as the heart use the malate-aspartate shuttle (MAS), where malate delivers NADH electrons to the mitochondrial complex I (yielding 2.25 ATP/NADH). The MAS includes cytosolic and mitochondrial transamination reactions that generate glutamate and aspartate (respectively), which are then exchanged between compartments. These reactions are catalyzed by glutamic-oxaloacetic transaminases 1 and 2 (Got1 and 2).

In addition to mitochondrial biogenesis and oxidative capacity, there are other pathways regulated by PGC-1α1 that contribute to muscle performance[1]. These include angiogenesis[12,13] and neuromuscular communication[14], among other. In addition, PGC-1α1 regulates a muscle to brain crosstalk, activated by aerobic exercise training, with a protective effect in the context of stress-induced depression[15]. This mechanism relies on increasing the expression of several kynurenine aminotransferases (Kats) in muscle, that clear the neurotoxic tryptophan metabolite kynurenine (Kyn) from circulation, and prevent its accumulation in the brain. Kyn accumulation in the CNS has been related to several mental health disorders[15–18]. Muscle Kats convert Kyn into kynurenic acid (Kyna)[15], which in turn increases adipose tissue energy expenditure and promotes an anti-inflammatory phenotype of the resident immune cells[19]. Both these biological activities of Kyna are mediated, at least in part, by GPR35 activation[19].

Here we show that PGC-1α1 induces the expression of glycolysis and MAS genes in skeletal muscle, thus increasing the energy efficiency of glucose oxidation. In addition, we show that this gene network includes both Kat and Got transaminases. This allows trained muscle to use Kyn catabolism to support energy production. Inhibition of Kat function, using gene silencing or chemical inhibitors (such as carbidopa) reduce myotube oxidative capacity and mouse exercise performance (respectively). Together, our data provide an integrated mechanism that explains how training can promote muscle neurotoxin detoxification while promoting adaptations to fatigue-resistance in the context of endurance exercise.

## Results and discussion

**PGC-1α1 induces MAS genes in skeletal muscle**. To discover PGC-1α1-regulated hubs in skeletal muscle we used a multi-omics data analysis approach. To this end, we analyzed muscle transcriptomics data obtained from mck-PGC-1α1 mice (Supplementary Data 1) together with metabolomics data reported for another mouse model of chronic PGC-1α elevation in skeletal muscle (HSA-PGC-1α-b)[20] (Fig. 1a). The results of this analysis suggested that PGC-1α1 positively regulates genes involved in the biosynthesis and metabolism of aspartate in skeletal muscle (Fig. 1b, c and Supplementary Fig. 1a). By quantifying the levels of these metabolites, we could confirm that aspartate and glutamate are indeed elevated in skeletal muscle of mck-PGC-1α1 mice when compared to wild-type (wt) controls (Fig. 1d). The same was observed in the muscle metabolomics data reported for the HSA-PGC-1α-b mice[20]. Aspartate and glutamate are intimately connected through glutamic-oxaloacetic transaminase 2 (GOT2). This transaminase is a crucial enzyme in aspartate biosynthesis[21,22] as it catalyzes the synthesis of aspartate from oxaloacetate and glutamate (Fig. 1e). Interestingly, GOT2 is also known as kynurenine aminotransferase 4 (KAT4), which we have previously shown to be regulated by PGC-1α1[15]. KAT enzymes use kynurenine (Kyn) and α-ketoglutarate to generate kynurenic acid (Kyna) and glutamate[23,24] (Supplementary Fig. 1b). Skeletal muscle PGC-1α1 increases the expression of Kat enzymes thereby enhancing Kyn to Kyna conversion[15]. This has benefits for mental health in the context of stress-induced depression, as Kyn accumulation in the CNS has neurotoxic effects[17]. In addition, the elevation in circulating Kyna levels regulates adipose tissue energy homeostasis[19].

**Muscle MAS gene clusters include Got and Kat transaminases**. Given that the reactions catalyzed by GOT2/KAT4 are involved in metabolic pathways, such as aspartate biosynthesis and kynurenine degradation[21–24], we next aimed to determine whether all KAT enzymes share a specific biological function in skeletal muscle. To this end, we used skeletal muscle expression data from isogenic but diverse mouse strains (BXD type). We then clustered Kat-associated gene modules by co-variation followed by prediction of biological function by gene ontology (Supplementary Fig. 2a, b). Interestingly, we observed that each *Kat* gene cluster includes several genes involved in the regulation of mitochondrial function and metabolism (Supplementary Fig. 2c–e). Given this biological association, we next aimed to find overlapping molecular processes between the *Kat*-associated networks. We then found that *Kat1* and *Got2/Kat4* strongly associate with genes regulating the MAS (Fig. 1f, g). This association could also be observed in human skeletal muscle using data that specializes in finding interacting genes that shape tissue-specific functions[25] (Supplementary Fig. 2f). Given that multi-protein network interactions are crucial for several biological processes across metazoans[26], we then used a composition of protein assemblies to confirm multiprotein networks associated with the *Kat1-Got2/Kat4* and malate-aspartate cluster. This revealed a conserved interacting protein network involved in glycolysis and pyruvate metabolism (Supplementary Fig. 3).

The MAS is a biochemical shuttle used by the heart and liver to transport glycolysis-derived electrons across the inner membrane of the mitochondria for oxidative phosphorylation[27] (Fig. 1k). Skeletal muscle has been described to use a different, less efficient shuttle mediated by G3P[27]. Importantly, GOT2/KAT4, glutamate,

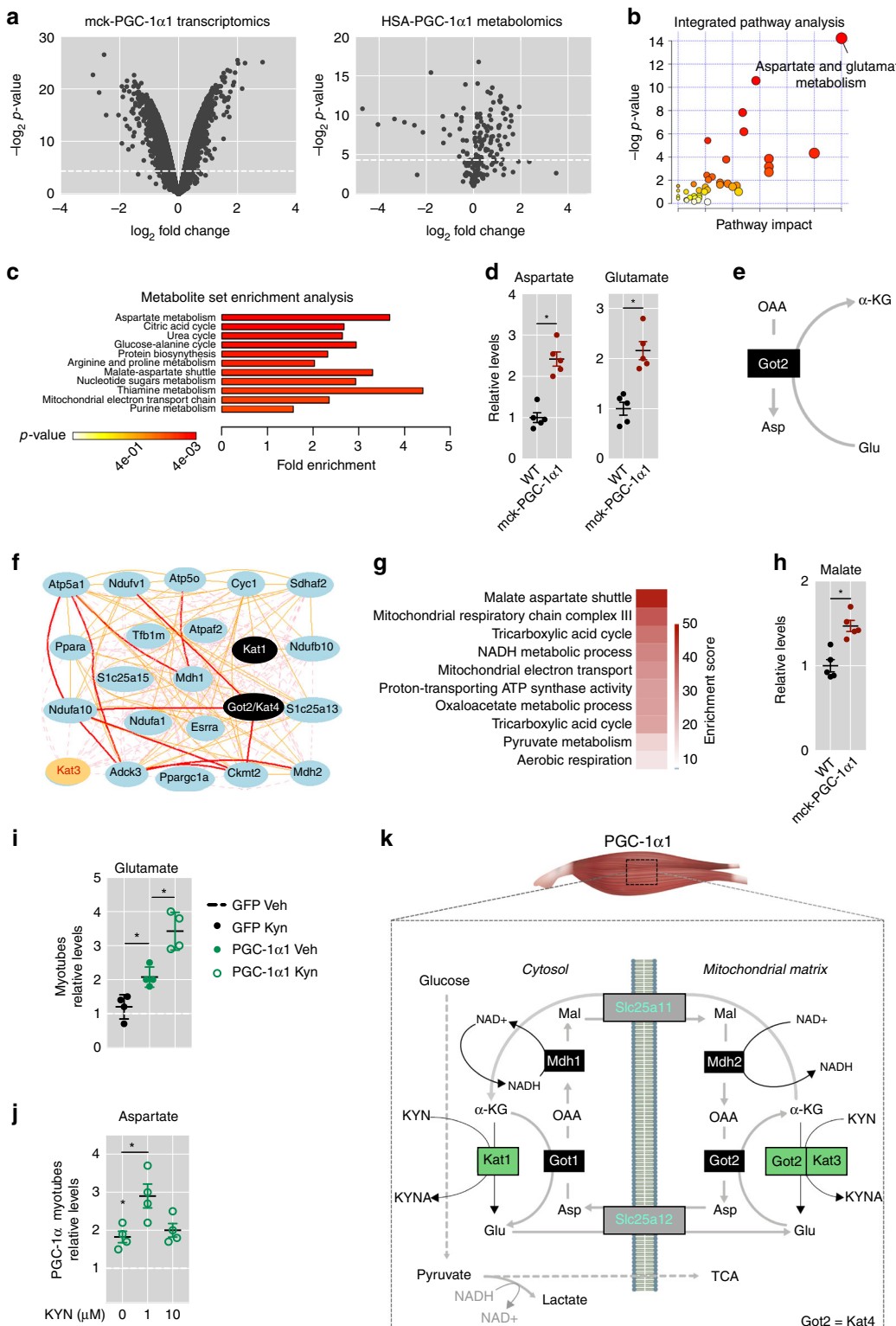

**Fig. 1** Regulatory hubs under PGC-1α1 control in skeletal muscle. **a** Transcriptomics and metabolomics analysis of mck-PGC-1α1 skeletal muscle. **b** Integrated pathway analysis of mck-PGC-1α1 transcriptomics and metabolomics. **c** Metabolite enrichment analysis of mck-PGC-1α1. **d** Relative levels of aspartate and glutamate in skeletal muscle of wt and mck-PGC-1α1 mice ($n = 5$). **e** Schematic representation of the role of GOT2 in aspartate synthesis. **f** Overlapping *Kat*-associated networks. **g** Functional annotation of the *Kat*-associated network. **h** Relative levels of malate in skeletal muscle of wt and mck-PGC-1α1 mice ($n = 5$). **i** Relative levels of glutamate in primary myotubes transduced with *Gfp* control or *Pgc-1α1* adenovirus and supplemented with 1 μM kynurenine (Kyn) ($n = 4$) **j** Relative levels of aspartate in primary myotubes transduced with *Pgc-1α1* adenovirus treated with 0, 1 or 10 μM Kyn ($n = 4$). **k** Schematic representation of how skeletal muscle PGC-1α1 integrates Kyn metabolism with aspartate biosynthesis and the malate-aspartate shuttle. Bars depict mean values and error bars indicate SEM. Unpaired, two-tailed student's *t*-test was used when two groups were compared, and one-way analysis of variance (ANOVA) followed by Fisher's least significance difference (LSD) test for *post hoc* comparisons were used to compare multiple groups, *$p < 0.05$

and aspartate are integral parts of the MAS. Within this shuttle, aspartate exported from the mitochondrial matrix is used as the precursor for malate, which in turn will be exchanged with α-ketoglutarate through SLC25A11 (Fig. 1k). In line with this, we found that the levels of malate are also increased in mck-PGC-1α1 muscle (Fig. 1h), showing a strong correlation with the levels of aspartate in skeletal muscle ($r > 0.5$) (Supplementary Fig. 4a). In agreement with the higher levels of these metabolites, we observed that skeletal muscle of mck-PGC-1α1 mice shows a robust increase in the expression of genes coding for all the constituents of the MAS (Supplementary Fig. 4b).

**PGC-1α1 and Kyn increase glutamate and aspartate levels**. To test if these effects could be observed in a cell-autonomous manner, we used primary myotube cultures with or without ectopic *Pgc-1α1* expression. We first determined if, as suggested by our bioinformatics analysis (Supplementary Fig. 3), PGC-1α1 would increase glycolytic flux in cultured myotubes by using cellular respirometry. Indeed, *Pgc-1α1* expression in myotubes increased ECAR levels at baseline, which were further increased by the addition of glucose, and further by the inhibition of mitochondrial respiration using oligomycin (Supplementary Fig. 4c). ECAR values returned to baseline when 2-deoxyglucose was added to inhibit glycolysis (Supplementary Fig. 4c). In addition, we found that increasing *Pgc-1α1* levels in myotubes is sufficient to increase glutamate levels (Fig. 1i) and the expression of MAS genes (Supplementary Fig. 5a), thus placing PGC-1α1 as a master regulator of the MAS in skeletal muscle. Surprisingly, we found that Kyn supplementation to myotubes further increased glutamate levels but only in the presence of *Pgc-1α1* (Fig. 1i). In agreement, treating PGC-1α1 myotubes with Kyn robustly elevated aspartate levels (Fig. 1j). Under these conditions, Kyn and PGC-1α1 induced the expression of MAS genes (Supplementary Fig. 5a), of several transcription factors associated with PGC-1α1 function (Supplementary Fig. 5b), and of mitochondrial genes (Supplementary Fig. 5c). As a consequence, supplementing *Pgc-1α1*-transduced myotubes with Kyn further elevated cellular respiration (Supplementary Fig. 5d, e). Importantly, Kyn-induced basal respiration was completely abolished by the addition of the ATP-synthase inhibitor oligomycin, indicating that the observed increased in OCR is completely coupled to ATP production (Supplementary Fig. 5e). These data indicate that PGC-1α1 confers myotubes with the ability to use Kyn to support the MAS, mitochondrial respiration, and energy production (Fig. 1k).

**PGC-1α1 reroutes Kyn catabolism to aspartate biosynthesis**. To confirm the role of muscle PGC-1α1 in rerouting Kyn for MAS support and energy production in vivo, we treated mck-PGC-1α1 mice with a single dose of Kyn. Again, we observed that Kyn was able to elevate the expression of genes involved in energy metabolism and MAS, but only in the mck-PGC-1α1 animals (Fig. 2a, b and Supplementary Fig. 6a, b). The fact that at 1 μM Kyn does not seem to have obvious effects on wt gastrocnemius (used in these experiments) could indicate that the lower levels of PGC-1α1 in these muscles (compared to more oxidative ones such as the soleus) are limiting for the activation of the KAT/MAS components (as previously suggested)[15]. Of note, transgenic expression of *Pgc-1α1* in the mck-PGC-1α mice, elevates overall muscle PGC-1α1 to the levels observed in wt soleus[5]. Also at the protein level, mck-PGC-1α1 mice showed increased expression of MAS transporters (SLC25A11 and SLC25A12; Fig. 2c and Supplementary Fig. 7). In agreement, skeletal muscle of mck-PGC-1α1 mice had elevated levels of glutamate, aspartate, and malate, which further increased with Kyn treatment (Fig. 2d). Although the MAS transports NADH generated during glycolysis directly

into the mitochondrial matrix, we also observed an increase in *Pdha1* mRNA expression upon acute Kyn administration (Supplementary Fig. 6b), which could be indicative of increased pyruvate utilization. To assess if Kyn affects PDH protein levels and/ or activation we determined the p-PDH/PDH ratio by immunoblotting and no significant difference was detected (Supplementary Fig. 6c). These results indicate that activation of the MAS by Kyn can increase ATP production by moving glycolysis-derived NADH from the cytosol into the mitochondrial electron transport chain, without changing pyruvate utilization.

**PGC-1α deletion renders muscle susceptible to Kyn toxicity**. Using a mouse model with skeletal muscle-specific deletion of *Pgc-1α1* (MKO-PGC-1α[14]), we observed that these knockout mice do not have the ability to use Kyn to support bioenergetics. Indeed, MKO-PGC-1α muscle has reduced baseline levels of genes involved in energy metabolism (including MAS genes), which often got further reduced by Kyn administration (Fig. 2e and Supplementary Fig. 6d–f). This was also translated into reduced aspartate levels in skeletal muscle that did not change after Kyn treatment (Fig. 2f). Interestingly, myotubes with genetic deletion of *Pgc-1α1* showed decreased respiration after Kyn treatment (Fig. 2g and Supplementary Fig. 6g–h). Strikingly, Kyn treatment reduced basal respiration to levels that were no longer affected by oligomycin treatment (Fig. 2g), indicating that these conditions lead to a severe compromise in ATP production. Treating wt myotubes with high concentration of Kyn decreased basal respiration (Fig. 2h and Supplementary Fig. 6i) and reduced aspartate levels (Fig. 2i). This suggests that, above a physiological threshold, Kyn-accumulation impairs aspartate biosynthesis and mitochondrial respiration. However, myotubes transduced with *Pgc-1α1* can withstand the effect of Kyn accumulation (10 μM) on maximal respiration (Supplementary Fig. 6j). Since muscle Kat expression is dependent on PGC-1α1[15], MKO-PGC-1α and wt myotubes will have a reduced capacity to convert Kyn into Kyna, thus allowing its degradation through the kynurenine pathway (KP)[17,18]. Interestingly, KP metabolites such as 3-hydroxykynurenine and 3-hydroxyanthranilic acid have been shown to impair energy metabolism by direct inhibition of mitochondrial respiration and ROS production[28,29].

**Contribution of MAS and fatty acids to myotube respiration**. Our results indicate that by allowing muscle to use Kyn and the MAS, PGC-1α1 enhances energy production from glycolysis. In agreement, Kyn supply to myotubes transduced with *Pgc-1α1* further increased ATP levels (Fig. 3a). This was accompanied by an elevation in the expression of glycolytic genes, which further increased after Kyn treatment both in vitro and in vivo (Fig. 3b, c). Although the mechanism for these PGC-1α1-dependent effects of Kyn on glycolytic gene expression remains unknown, it is tempting to speculate that Kyn to Kyna conversion could regulate the expression or activity of a transcription factor yet to be identified. To date, the only transcription factor known to be activated by both Kyn and Kyna is the Aryl hydrocarbon receptor (AHR), which has recently been shown to directly regulate glycolytic genes in type 1 regulatory T cells[30]. Activation of the MAS in skeletal muscle could be in addition to or in replacement of the G3PS[26]. Interestingly, PGC-1α1 does not influence the expression of G3PS enzymes in vivo, in vitro or after Kyn treatment (Fig. 3d and Supplementary Fig. 8a, b). To determine the functional effects of inducing the MAS in skeletal muscle, we then inhibited fatty acid oxidation by using a carnitine palmytoyltransferase-1b (CPT1B) chemical inhibitor (Etomoxir). This reduced basal and maximal respiration in wt myotubes by 64 and 75% (respectively), whereas in myotubes overexpressing *Pgc-1α1* the effect of

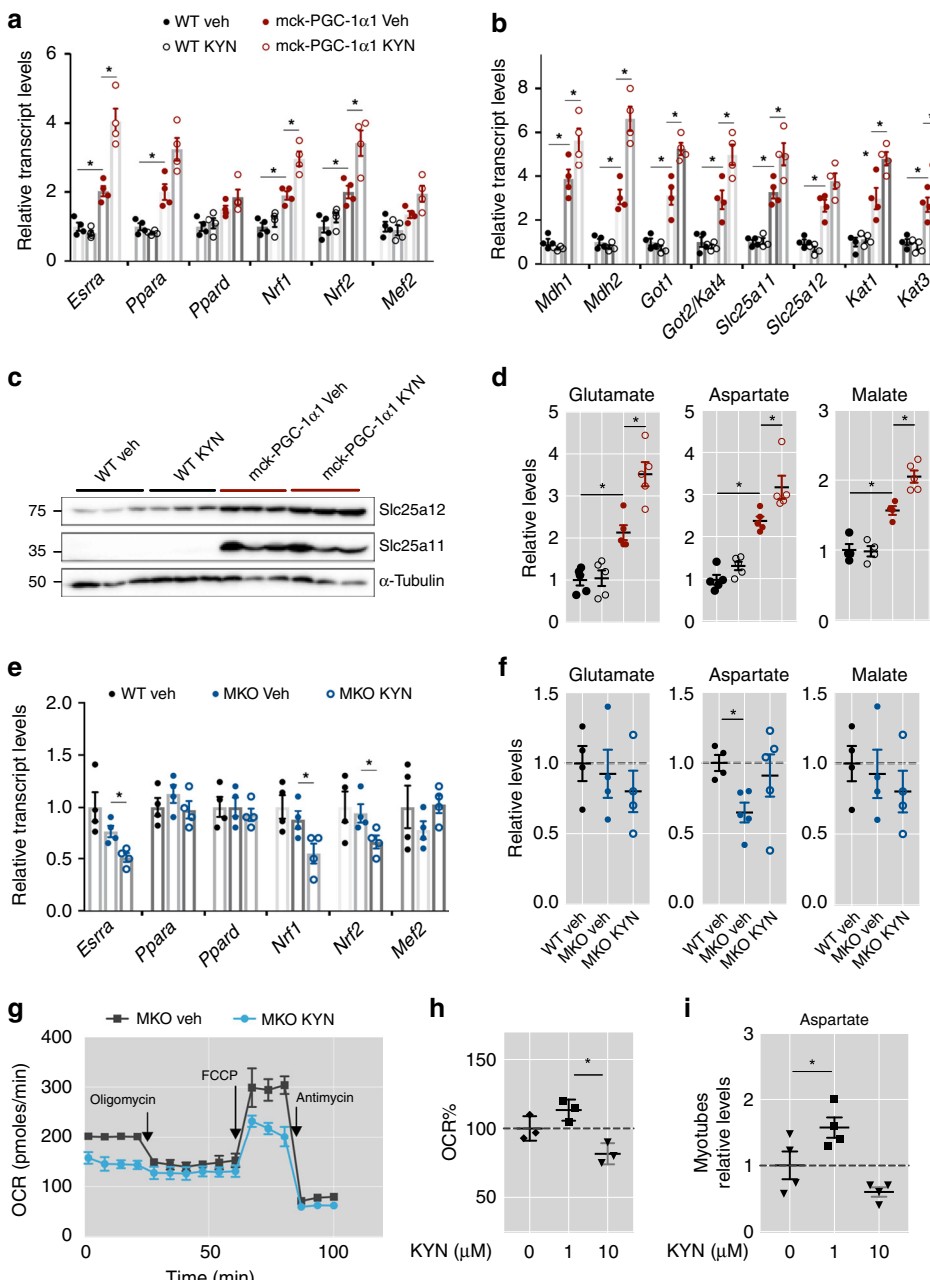

**Fig. 2** PGC-1α1 reroutes kynurenine catabolism to support aspartate biosynthesis. **a** Relative transcript levels of mitochondrial related genes in skeletal muscle of wt and mck-PGC-1α1 mice with a single intraperitoneal dose of kynurenine (Kyn, 2.5 mg/kg) ($n = 4$). **b** Relative transcript expression of genes involved in malate-aspartate metabolism in skeletal muscle of wt and mck-PGC-1α1 mice treated as in **a**. **c** Protein levels of the malate-aspartate shuttle constituents SLC25A11 and SLC25A12 in skeletal muscle of wt and mck-PGC-1α1 mice with a single intraperitoneal dose of Kyn. Uncropped western blots are found as Supplementary Fig. 7. **d** Relative levels of glutamate, aspartate and malate in skeletal muscle of mck-PGC-1α1 mice treated as in **a**. **e** Relative transcript levels of mitochondrial related genes in skeletal muscle of wt and MKO-PGC-1α mice with a single intraperitoneal dose of Kyn ($n = 4$). **f** Relative levels of glutamate, aspartate and malate in skeletal muscle of wt and MKO-PGC-1α mice with a single intraperitoneal dose of Kyn ($n = 4$). **g** Extracellular Flux Analysis (Seahorse[TM]) of cellular respiration in primary myotubes from MKO-PGC-1α supplemented with PBS (veh) or with 1 μM Kyn ($n = 4$). **h** Percentage of basal oxygen consumption rate in primary myotubes supplemented with 1 and 10 μM of kynurenine (Kyn) ($n = 4$). **i** Relative levels of aspartate in primary myotubes transduced with *Pgc-1α1* adenovirus and treated as in **h** ($n = 4$). Bars depict mean values and error bars indicate SEM. Unpaired, two-tailed student's t-test was used when two groups were compared, and one-way analysis of variance (ANOVA) followed by Fisher's least significance difference (LSD) test for *post hoc* comparisons were used to compare multiple groups, *$p < 0.05$

Etoxomir was significantly blunted (with a reduction of 35 % in basal and 50 % maximal respiration; Fig. 3e). In fact, treating myotubes overexpressing *Pgc-1α1* with Etomoxir further increased aspartate levels (Fig. 3f). Given that PGC-1α1 elevates the expression of *Cpt1b*, we checked if this could affect the experimental outcome but found no differences in its expression

after Etomoxir treatment (Supplementary Fig. 8c). This indicates that myotubes overexpressing PGC-1α1 acquire the plasticity to enhance glucose-malate-aspartate metabolism even when fatty acid oxidation is reduced. To further assess the functional contribution of the MAS in myotubes, we used the aminotransferase inhibitor aminooxyacetate (AOA,100 μM), that works as a

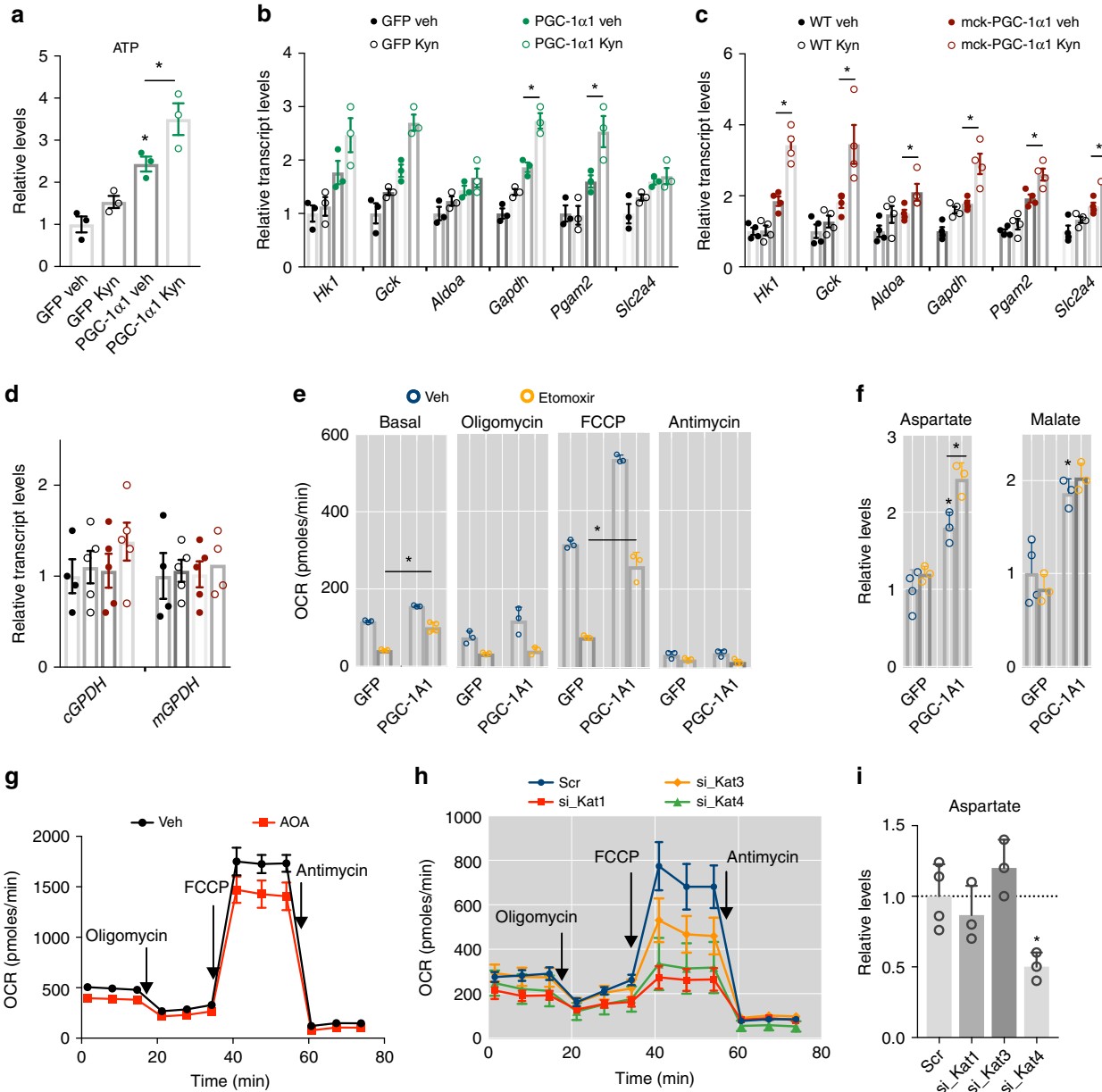

**Fig. 3** The bioenergetic role of Kat/malate-aspartate shuttle integration in skeletal muscle. **a** Relative levels of ATP in primary myotubes transduced with *Gfp* control or *Pgc-1α1* adenovirus supplemented with 1 μM kynurenine (Kyn) ($n = 4$). **b** Relative transcript levels of glycolytic genes in primary myotubes transduced with *Gfp* control or *Pgc-1α1* adenovirus and supplemented with 1 μM Kyn ($n = 4$). **c** Relative transcript levels of glycolytic genes in skeletal muscle of wt and mck-PGC-1α1 mice with a single intraperitoneal dose of Kyn (2.5 mg/kg) ($n = 4$). **d** Relative transcript levels of glycerol-3-phosphate shuttle genes in skeletal muscle of wt and mck-PGC-1α1 mice treated as in **c**. **e** Cellular respiration measured by Extracellular Flux Analysis (Seahorse™) in primary myotubes transduced with *Gfp* control or *Pgc-1α1* adenovirus and supplemented with 50 μM Etomoxir for 1 h ($n = 4$). **f** Relative levels of aspartate and malate in primary myotubes transduced with *Gfp* control or *Pgc-1α1* adenovirus and supplemented with 50 μM Etomoxir for 1 h ($n = 4$). **g** Cellular respiration measured as in **e** of primary myotubes supplemented with 100 μM aminooxyacetate (AOA) for 1 h ($n = 4$). **h** Cellular respiration measured as in **e** of primary myotubes transfected with scrambled siRNA, or siRNAs for *Kat1*, *Kat3* or *Got2/Kat4* ($n = 4$). **i** Relative levels of aspartate in primary myotubes transfected as in **h**. Bars depict mean values and error bars indicate SEM. Unpaired, two-tailed student's *t*-test was used when two groups were compared, and one-way analysis of variance (ANOVA) followed by Fisher's least significance difference (LSD) test for post hoc comparisons were used to compare multiple groups, $*p < 0.05$

chemical inhibitor of this shuttle[31]. Acute AOA treatment reduced overall cellular respiration in myotubes (Fig. 3g) without affecting ECAR (Supplementary Fig. 8e). In parallel, by using transient siRNA transfections we reduced SLC25A12 and SLC25A11 expression by 44 and 34%, respectively (Supplementary Fig. 8f). Under these conditions we did not observe any effects on OCR, even when Kyn was added (Supplementary Fig. 8g). Interestingly, *Slc25a12* silencing elevated ECAR values at

baseline with little effect of Kyn addition, whereas *Slc25a11* silencing had a more robust effect on ECAR values upon Kyn addition (Supplementary Fig. 8h). In a similar fashion, *Slc25a12* knockdown resulted in significantly elevated aspartate levels that were not affected by Kyn, whereas lowering *Slc25a11* expression had no effect (despite a trend towards elevated aspartate after Kyn) (Supplementary Fig. 8i). Although the aspartate levels we observed are within the range previously reported in human

muscle[32], the increase in aspartate we saw upon myotube *Slc25a12* silencing is in contrast with other studies[33–35]. This could be due to the different levels of PGC-1α1 and Kat enzyme expression in the different cells used (e.g. mouse myotubes vs C2C12 myoblasts), as well as the presence of kynurenine and other factors in the culture media.

Given the contribution of Kyn and KATs to support the MAS, we next aimed to determine the individual functional effect of the different KAT enzymes. We observed that silencing *Kat1* and *Got2/Kat4* expression robustly decreased basal and maximal respiration (Fig. 3h and Supplementary Fig. 8d), while increasing ECAR (Supplementary Fig. 8j). As expected, aspartate levels were reduced upon *Got2/Kat4* silencing (Fig. 3i). Together, these data indicate that interfering with the MAS at different points can have different consequences such as decreased oxygen consumption, increased glycolysis-derived lactate production, or both. This interdependence on cytosolic and mitochondrial components is not surprising. Indeed, it has been shown that whereas mck-PGC-1α1 muscle is more efficient than wt at using pyruvate, isolated mitochondria from the same animals are not[36].

**Carbidopa reduces mouse exercise performance and strength**. We next evaluated the effects of Kat inhibition using carbidopa, which irreversibly inactivates the Kat cofactor pyridoxal-5′-phosphate (PLP)[37]. Interestingly, carbidopa is commonly used as a dopamine decarboxylase inhibitor, which, in combination with levodopa, represents a common therapy for pathologies such as Parkinson's disease and restless legs syndrome. However, since carbidopa is only used in combination with levodopa, there are limited reports addressing the effects of carbidopa alone on peripheral tissues[38]. In line with our previous observations, treating myotubes with carbidopa resulted in reduced maximal respiration (Fig. 4a). To bypass the contribution of anaerobic glycolysis, we incubated the myotubes with only pyruvate and observed that carbidopa-mediated inhibition of respiration was exacerbated in this condition (Fig. 4a). In fact, myotubes in pyruvate that were treated with both carbidopa and Etomoxir showed a drastic reduction in basal and maximal respiration (Fig. 4a). Confirming our previous observations (Fig. 3), these effects were less pronounced in myotubes overexpressing *Pgc-1α1* (Fig. 4a). Accordingly, carbidopa treatment in myotubes reduced aspartate and malate levels (Fig. 4b), and the expression of genes involved in energy metabolism (Supplementary Fig. 9a, b). Interestingly, the effects of carbidopa on basal and maximal cellular respiration could be partially rescued by aspartate treatment (Fig. 4c).

To verify the impact of skeletal muscle Kat inhibition upon a physiological-challenge, we evaluated the effects of sub-chronic carbidopa administration in C57bl/6J mice using an exercise performance test. We found that animals treated with carbidopa run 30% less than controls in an exercise performance test (Fig. 4d and Supplementary Fig. 9c) and had reduced expression of genes involved in energy metabolism in skeletal muscle (Supplementary Fig. 9d). Importantly, the deleterious effects of carbidopa could be rescued by aspartate treatment (Fig. 4d and Supplementary Fig. 9c). In addition, using isolated flexor digitorum brevis muscle, we observed that carbidopa pre-treatment elicited a decrease in relative contractile force, in a fatigue-induction protocol (Fig. 4e). Interestingly, there are several heart and skeletal muscle-related side effects and adverse reactions associated with carbidopa[38]. Our observations suggest that the effects of carbidopa on muscle metabolism could relate to some of these adverse effects and that aspartate supplementation could have clinical applications.

Finally, to determine if the MAS is part of the physiological adaptation to exercise training, we analyzed malate and aspartate metabolism in murine and human skeletal muscle after chronic exercise interventions. We found that in mice, endurance exercise training induces the expression of glycolysis and MAS-related genes as well as malate and aspartate metabolite levels (Fig. 5a, b and Supplementary Fig. 10a–c). Importantly, the effects of exercise on MAS-related genes were lost in muscle-specific *Pgc-1α* knockouts following both acute and chronic exercise interventions (Fig. 5d and Supplementary Fig. 10d). The effects of exercise training could also be observed in human skeletal muscle (Fig. 5e, f and Supplementary Fig. 10e, f). Protein levels of MAS components could also be found elevated by proteomics analysis or trained vs untrained murine and human muscle (Fig. 5c, g, respectively)[39–42].

## Conclusions

To date, at least three different mouse models of elevated *Pgc-1α* expression in skeletal muscle have been reported, but not always generated overlapping results (mck-PGC-1α[5]; PGC-1α TRE (+)[43]; HSA-PGC-1α-b[44]). The three models use different strategies to elevate PGC-1α in muscle. Mck- and HSA-driven *Pgc-1α* expression is already activated at birth and remains on during the life of the mice. These were the models used to generate the transcriptomics (this study) and metabolomics[20] data, respectively. Indeed, the metabolomics data obtained with the HSA-PGC-1α-b mice, already highlighted the elevation in MAS metabolites[20]. The PGC-1α TRE (+) model is an inducible model (mice were kept on doxycycline which represses transgene expression until it was removed from the diet, at about 5–6 weeks of age). Evidence of differences between the models is reflected on exercise performance, GLUT4 levels, glucose tolerance, insulin sensitivity, among other. Importantly, PGC-1α TRE (+) mice show a reduction in exercise capacity[43], whereas the mck-PGC-1α mouse has increased exercise performance[45]. Interestingly, both models show increased muscle glycogen content at rest but the mck-PGC-1α can use it during exercise[46] whereas the PGC-1α TRE (+) cannot[43]. Importantly, it has been shown that that human type I skeletal muscle fibers (rich in PGC-1α1) exhibit higher levels of MAS enzymes, whereas type II fibers have more cGPDH[47]. In the same publication, the authors show that after endurance training both fiber types have higher levels of MAS enzymes.

Taken together, our results uncover a pathway regulated by PGC-1α1 (Fig. 1k), which allows trained skeletal muscle to use Kyn to support aspartate biosynthesis and mitochondrial function. These findings have implications on our understanding of how aerobic training renders skeletal muscle resistant to fatigue, by making glucose utilization more efficient. This delays the switch to fatty acid oxidation, a much more oxygen-demanding process, which would precipitate reaching maximal aerobic capacity. In addition, the observation that carbidopa inhibits this process, and reduces the oxidative capacity of muscle, could have important clinical implications[38].

## Methods

**Animal experiments**. Mck-PGC-1α1 and MKO-PGC-1α animals (all on C57bl/6J background), a kind gift from Dr. Bruce Spiegelman (Harvard Medical School, Boston, MA), have been previously described[5,13]. Wild-type mice (C57bl/6J) were procured from Janvier Labs (France). Mice were housed in plastic cages (3–5 per cage) at 24 ± 1 °C, 12/12 h controlled light conditions with ad libitum access to water and food. All mice were male and 10–14 weeks of age. All experiments were approved by the regional animal ethics Committee of Northern Stockholm, Sweden or of Basel-Stadt, Switzerland.

**Chemical treatments**. Differentiated myotubes were treated for 4 h with L-Kynurenine (Kyn) sulfate salt (Sigma-Aldrich; 1 μM). When stated, differentiated

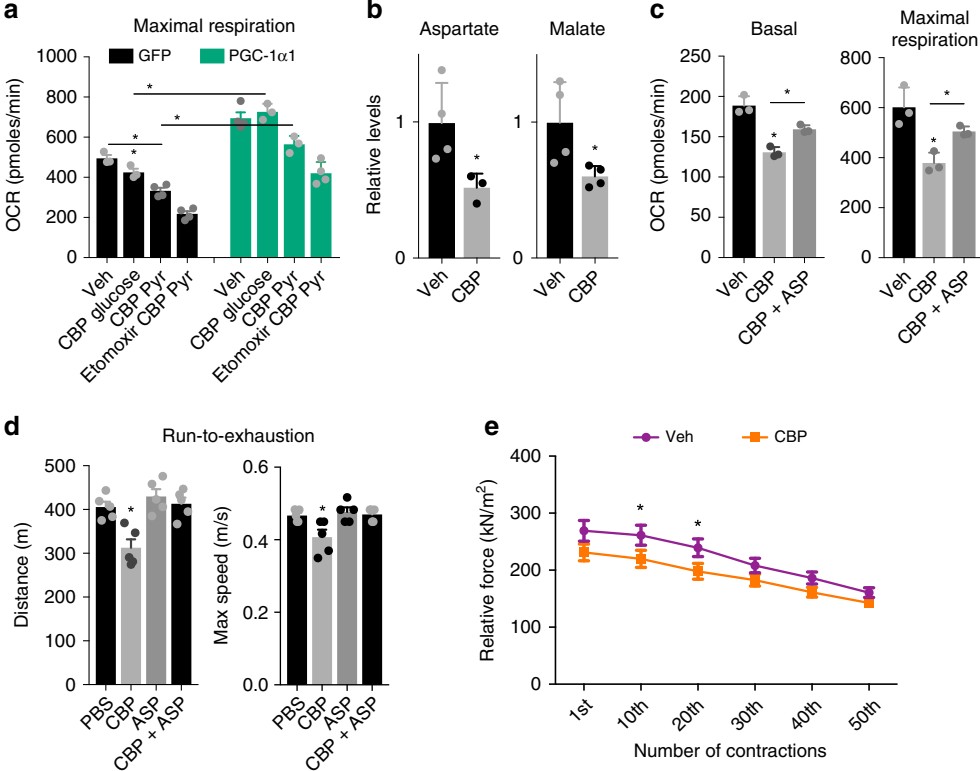

**Fig. 4** Kat-inhibition by carbidopa impairs malate-aspartate metabolism in skeletal muscle. **a** Maximal respiration measured by Extracellular Flux Analysis (Seahorse[TM]), in the presence of Glucose or Pyruvate (Pyr), in primary myotubes transduced with *Gfp* control or *Pgc-1α1* adenovirus and supplemented with 40 μM carbidopa (CBP) for 24 h and/or 50 μM Etomoxir for 1 h ($n = 4$). **b** Relative levels of aspartate and malate in primary myotubes supplemented with 40 μM carbidopa for 24 h. **c** Basal and maximal respiration of primary myotubes supplemented with 40 μM carbidopa for 24 h or 40 μM carbidopa for 24 h followed by 100 μM aspartate (ASP) for 1 h. **d** Mouse exercise performance test shown as distance and maximal speed after 4 days of intraperitoneal injection of PBS, carbidopa, aspartate or carbidopa + aspartate ($n = 5$). **e** Tetanic force was measured at 70 Hz force with 350 ms tetani at 2 s intervals for 50 contractions in skeletal muscle of mice after 4 days of intraperitoneal injection of PBS or carbidopa ($n = 4$). Bars depict mean values and error bars indicate SEM. Unpaired, two-tailed student's *t*-test was used when two groups were compared, and one-way analysis of variance (ANOVA) followed by Fisher's least significance difference (LSD) test for *post hoc* comparisons were used to compare multiple groups, *$p < 0.05$

myotubes were treated with different concentrations of Kyn (1 and 10 μM) for 4 h. C57bl/6J, mck-PGC-1α1 and MKO-PGC-1α mice received intraperitoneal injections of Kyn (2.5 mg/Kg) or PBS 3–4 h before sacrifice.

Differentiated myotubes were treated for 24 h with L-carbidopa (CBP) (Sigma-Aldrich; 40 μM). When stated, myotubes where supplemented after CBP treatment with L-Aspartate (Sigma-Aldrich; 100 μM) 1 h before cell collection or Extracellular flux analysis assay. C57bl/6J mice received CBP (25 mg/Kg) or PBS intraperitoneally once per day for 4 days. When stated, L-Aspartate (12.5 mg/Kg) was intraperitoneally injected alone or together with CBP injections for 4 days.

Differentiated myotubes were treated for 1 h with Etomoxir (Sigma-Aldrich; 50 μM). When stated, myotubes where supplemented, after CBP treatment with Etomoxir 1 h before cell collection or Extracellular flux analysis assay. Aminooxyacetate (AOA) was used at 100 μM for 1 h before measurement. Differentiated myotubes were transfected using Lipofectamine RNAMax (Life Technologies) with 25 nM of siRNA for *Kat1* (Thermo Fisher Scientific siRNA ID # 181312; AGU AGU AGC UUG GAC GAA A), siRNA for *Kat3* (Thermo Fisher Scientific siRNA ID # 171263; GCC CUA CUU CAC CUG AUC G), siRNA for *Kat4* (Thermo Fisher Scientific siRNA ID # 158792; CAA UAG CGA UGA UAC UGG G), siRNA for *Slc25a11* (Dharmacon siRNA ID # 67863; UGG CCG GAU GGA CGG GAA A; GGA CUA GUG UGC CAG GCU U; AAA CUA GGA UCC AGA AUA A; AGU UAG GAU UGC GAC GGA A), siRNA for *Slc25a12* (Dharmacon siRNA ID # 78830; GCU CCA AGA UUG CGA GAA A; GAU ACA AGG CAG AGC GAA A; GUU UAA GUC UCC UAG CGU A; ACA UGG AGC UUG UUC GAA A) or a scrambled control (Life Technologies).

**Extracellular flux analysis (Seahorse®) assays.** Cells were differentiated and treated as described above. On the 4th day of differentiation, mitochondrial oxidative phosphorylation from mouse differentiated myotubes was analyzed using extracellular flux analysis (XF24; Seahorse Biosciences) in DMEM medium (pH 7.4; Sigma-Aldrich). Unless otherwise stated, the Mitochondrial Stress Test assay was performed with 1 mM pyruvate and 25 mM glucose (Sigma-Aldrich). Baseline

oxygen consumption rates (OCR) and extracellular acidification rate (ECAR) were measured every 7th minute. Following baseline measurements, oligomycin (1 μM), Carbonyl cyanide-4 (trifluoromethoxy)phenylhydrazone (FCCP) (1 μM), and antimycin A (2 μM) were sequentially injected to measure OCR and ECAR. The glycolytic flux was measured using the Glycolytic Stress Test assay performed in DMEM medium (pH 7.4; Sigma-Aldrich). Baseline ECAR was measured every 7th minute. Following baseline measurements, glucose (10 mM), oligomycin (1 μM), and 2-deoxyglucose (100 mM) were sequentially injected followed by ECAR measurements.

**Analysis of gene expression.** Total RNA was isolated from cells or tissues using Isol-RNA Lysis Reagent (5 PRIME) according to manufacturer's instructions. One microgram of RNA was treated with Amplification Grade DNase I (Life Technologies) and from that, 500 ng of total RNA were used for cDNA preparation using the Applied Biosystem Reverse Transcription Kit (Life Technologies). Quantitative Real-Time PCR was performed in a ViiA 7 Real-Time PCR system thermal cycler with SYBR Green PCR Master Mix (both Applied Biosystems). Analysis of gene expression was performed using the ΔΔCt method and relative gene expression was normalized to hypoxanthine phosphoribosyl-transferase (*Hprt*) mRNA levels. Gene expression analyses were expressed as mRNA levels relative to controls. Primer sequences are found in Supplementary Data 2.

**Primary cell culture.** Primary mouse myoblasts were isolated from 2-weeks-old C57bl/6J or MKO-PGC-1α mice as previously described[4]. In brief, mouse hindlimb muscles were dissected out and digested to a homogenate in a collagenase/dispase solution (2.4 U/mL Dispase (Grade II) (Roche 295825); 1% Collagenase B (Roche 1088815); 2.5 mM CaCl2; in MilliQ water). Myoblasts were cultured in collagen-coated plates and maintained in Ham's F-10/DMEM media mixture (Thermo Fisher Scientific) supplemented with 20% FBS (Sigma-Aldrich), 1%

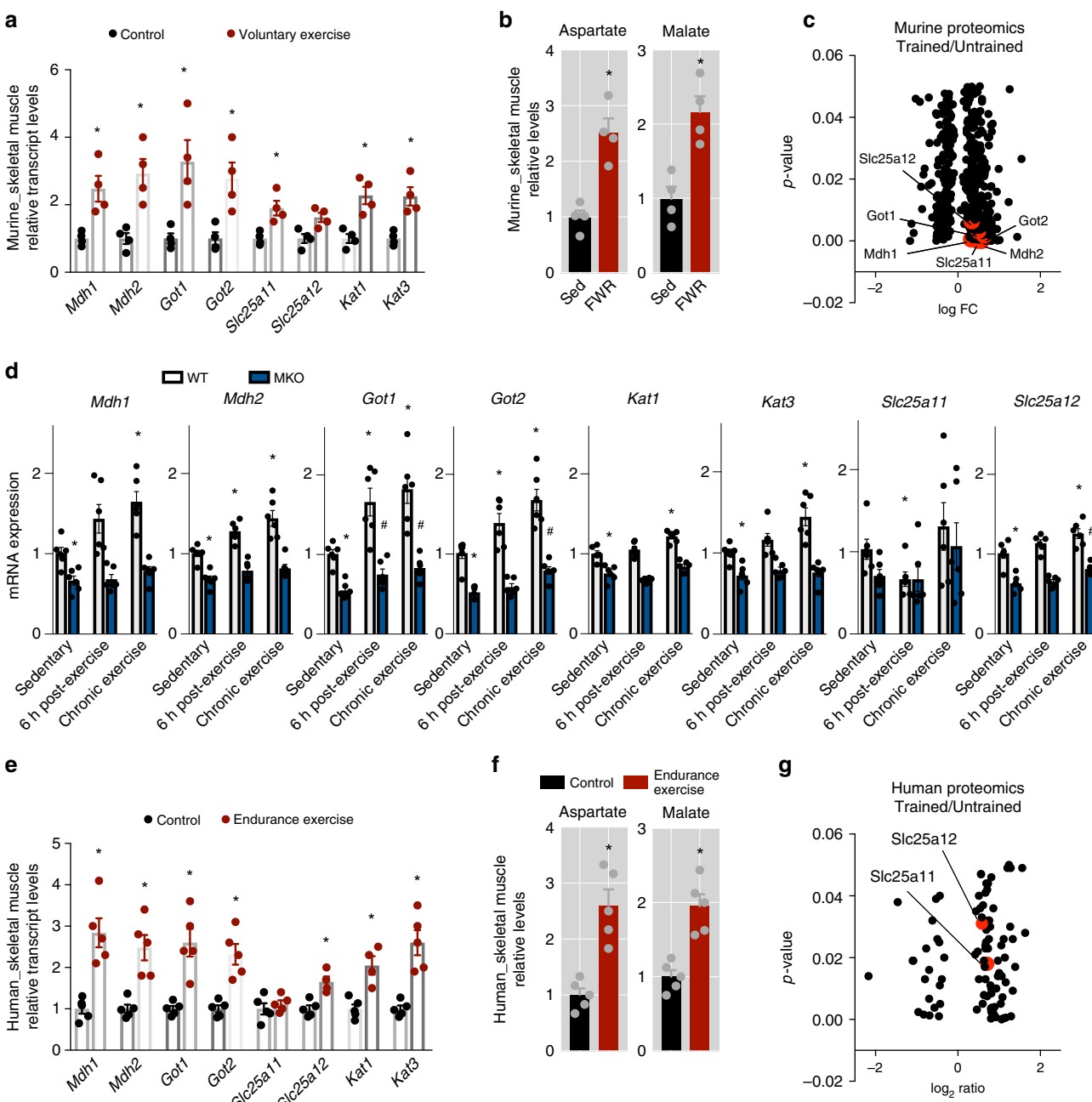

**Fig. 5** Endurance exercise effects in both murine and human skeletal muscle. **a** Relative transcript levels of genes involved in malate-aspartate metabolism in skeletal muscle of sedentary wild-type mice and wild-type mice with access to free-wheel running (FWR) for 8 weeks. Mice were sacrificed 12 h after the last bout of exercise ($n = 4$). **b** Relative levels of aspartate and malate in skeletal muscle of sedentary and wild-type mice with access to FWR ($n = 4$). **c** Compared proteomics from skeletal muscle of trained and untrained mice[15]. **d** Relative transcript levels of genes involved in malate-aspartate metabolism in skeletal muscle of sedentary, acutely and chronically exercised wt and MKO-PGC-1α mice ($n = 5$–6). **e** Expression of genes involved in malate-aspartate metabolism in skeletal muscle of human volunteers after endurance exercised ($n = 5$). **f** Relative levels of aspartate and malate in skeletal muscle of human volunteers after endurance exercise ($n = 5$). **g** Compared proteomics from skeletal muscle of human volunteers after endurance exercise[17,23,24]. Bars depict mean values and error bars indicate SEM. Unpaired, two-tailed student's $t$-test was used when two groups were compared, and one-way analysis of variance (ANOVA) followed by Fisher's least significance difference (LSD) test for *post hoc* comparisons were used to compare multiple groups, *$p < 0.05$

penicillin/streptomycin (Thermo Fisher Scientific), and 2.5 ng/mL basic fibroblast growth factor (Thermo Fisher Scientific). To induce differentiation into myotubes, cells were shifted to DMEM high glucose with pyruvate (Thermo Fisher Scientific) media supplemented with 5% horse serum (Thermo Fisher Scientific) and 1% penicillin/streptomycin (Thermo Fisher Scientific). When stated, 48 h after induction of differentiation, cells were transduced with an adenovirus to induce *Pgc-1α1* expression or a control adenovirus. Twelve hours after transduction medium was replaced with fresh differentiation medium and approximately 48 h post-transduction, cells were harvested.

**Adenovirus-mediated expression.** Adenovirus expressing *Gfp* and *Pgc-1α1* were generated by using the pAdTrack/pAdEasy system (Stratagene). For cell culture experiments, differentiated myotubes were transduced with an adenovirus at an MOI of 100 for 12 h. Cells were harvested 2 days after transduction.

**Western blot.** Cell or tissues lysates were obtained using RIPA buffer (50 mM tris pH 7.5, 150 mM NaCl, 1 mM EDTA, 1% (w/v) Triton-X-100, 0.5% (w/v) Na-deoxycholate, 0.1% (w/v) sodium dodecyl sulfate, 20 mM glycerol-2-phosphate,

5 mM sodium pyrophosphate) freshly supplemented with 1 mM DTT and 0.5 mM PMSF. Each protein extract was resolved by SDS polyacrylamide gel membranes (PAGE), and blots were incubated with the monoclonal antibodies anti-PDH-E1α (9H9AF5; Thermo Fisher Scientific) or anti-Slc25a12 (kind gift of Dr. Jorgina Satrústegui), or the polyclonal antibodies anti-phosphoPDH-E1α Ser[290] (AP1062; Calbiochem) or anti-Slc25a11 (AB80464; Abcam). Samples were normalized using monoclonal antibody anti-α-tubulin (T6199; Sigma-Aldrich). All antibodies were used at 1:1000, except anti-Slc25a12 that was used at 1:5000.

Uncropped and unprocessed scans of the westerns shown in Fig. 2c are found in Supplementary Fig. 7.

**Human endurance exercise.** Skeletal muscle biopsies from two groups of young adult male subjects were used. The first group ($n = 9$) was pursuing endurance training for a 150-km road cycling time trial ($11.8 \pm 4.8$ training hours per week), and the second group ($n = 8$) was recreationally active but not specifically participating in endurance training. Skeletal muscle biopsies were taken 12 h after exercise completion from *m. vastus lateralis* under local anesthesia using the Pajunk DeltaCut system (Pajunk, Geisingen, Germany). Approximately 40 mg of muscle tissue was collected in the rested conditions for each subject, snap frozen in liquid nitrogen, and stored at −80 °C until further analysis. The study was approved by the regional ethics committee and each subject provided written informed consent before participation.

**Free wheel running.** Nine-week-old C57bl/6J male mice were single-housed and divided into two groups: control and exercise. After an acclimation period of 5–6 days, the exercised group was given access to a running wheel with a counter that monitored revolutions during 8 weeks. Control group mice were housed in similar cages without running wheels. Only animals that had run more than 4 km/night were selected for subsequent experiments. *Gastrocnemius* muscles were collected at 12 hours after the last bout of exercise.

**Exercise performance.** Treadmill runners ($n = 6$) followed an acclimation protocol (day 1 running 5 m/min for 10 min; day 2 running 5 m/min for 10 min followed by 10 m/min for 5 min; day 3 running 5 m/min for 10 min followed by 10 m/min for 5 min and 15 m/min for 2 min). After acclimation, mice were placed on a motorized treadmill (Columbus Instruments, USA), with a slope of 5° and warmed-up for 5 min at 5 m/min. The speed was increased (2 m/min every 2 min) up to 35 m/min. Speeds varied between 10 and 25 m/min during the experiment to allow mice to keep up with the protocol. Mice ran for 1 h or until exhaustion. We considered exhaustion as the total incapacity to keep up with the running pace even when the speed was lowered to 15 m/min. *Gastrocnemius* muscles were collected 12 h after the running to exhaustion protocol was performed.

**Acute and chronic exercise.** Prior to the long endurance exercise exhaustion test or chronic exercise training, MKO-PGC-1α or wild-type littermate control mice were familiarized with treadmill running. The exhaustion test was performed at an inclination of 5° and started at 5 m/min for 5 min followed by 5 min at 8 m/min. Subsequently, velocity was increased by 2 m/min every 15 min until mice reached exhaustion (criteria described above). Mice were sacrificed 6 hours post-exhaustion and *quadriceps* muscles were removed and snap frozen in liquid nitrogen. Chronic exercise training was performed on a motorized treadmill for 4 weeks. Mice were trained 5 days/week for 1 h, progressively increasing the velocity throughout the training period (inclination of 5°). 18 hours after the last training session, mice were sacrificed and *quadriceps* muscles was removed and snap frozen in liquid nitrogen.

Muscle tissue from these exercise studies was homogenized in 1 mL TRIzol Reagent (Thermo Fisher Scientific) using FastPrep lysing matrix tubes (MP Biomedicals). RNA was isolated with the TRIzol extraction method and performed according to the manufacturer's protocol. One µg RNA was treated with Amplification Grade DNase I (Life Technologies) and reverse transcribed using the High-Capacity cDNA Reverse Transcription Kit (Applied Biosystems). Quantitative Real-Time PCR was performed on the QuantStudio 5 using Fast SYBR Green PCR Master Mix (both Applied Biosystems).

**Skeletal muscle force and fatigue measurement.** The flexor digitorum brevis (FDB) muscles from both hind paws of C57bl/6J mice after CBP or PBS injection were quickly excised under the microscope after the sacrifice. During the excision procedure and force measurements, the muscles were kept in a Tyrode solution containing (in mmol/L): 121 NaCl, 5 KCl, 1.8 $CaCl_2$, 0.4 $NaH_2PO_4$, 0.5 $MgCl_2$, 24 $NaHCO_3$, 0.1 EDTA, and 5.5 glucose. The Tyrode solution was gassed with 95% $O_2$-5% $CO_2$, giving a bath pH of 7.4. The proximal and distal tendons of FDB muscles were tied with nylon thread. Muscles were mounted between a force transducer and an adjustable holder (World Precision Instruments) in a 15 mL stimulation chamber. The chamber temperature was set at 31 °C with a water-jacketed circulation bath and the muscles were bathed in the Tyrode solution continuously gassed with 95% $O_2$-5% $CO_2$. Muscles were stimulated with supra-maximal current pulses (0.5 msec duration; 150% of current required for maximum force response) via plate electrodes lying parallel to the muscles. Muscles were set to the length at which tetanic force was maximum (optimal length $L_0$) and

were then allowed to recover for 15 min. $L_0$ was measured with a caliper and recorded. FDB muscles underwent a fatigue protocol consisting of 50 tetanic contractions (70 Hz, 300 ms train duration, 2 s interval duration). Electrically stimulated force production was expressed as specific force ($kN/m^2$). Muscle cross-sectional area (CSA) was assessed by dividing muscle mass by the product of muscle length and muscle density ($1.06 \, g/cm^3$). The muscle mass was determined after the experiments by cutting the major part of the tendons.

**Kit measurements.** Aspartate, Malate, and Glutamate quantification from mouse skeletal muscle samples were done using a commercially available kit (Abcam ab102512, ab83391, and ab138883, respectively), following manufacturer's instructions.

**Transcriptomic and metabolomic analysis.** We analyzed muscle transcriptomics (Supplementary Data 1) and metabolomics[20] data obtained from skeletal muscle of mck-PGC-1α1 mice. Metabolite set enrichment analysis was performed using the R package MetaboAnalystR[48,49]. Integrated metabolic pathway analysis of transcriptomic and metabolomics data was performed using the same R package.

**Proteomics analysis.** We analyzed publicly available data from muscle proteomics from murine[39] and from humans[40–42] obtained from skeletal muscle after exercise training.

**KAT-associated network.** To assess the KATs-associated network in murine skeletal muscle, we used RNAseq data from isogenic BDX mice (archived in http://www.GeneNetwork.org)[50]. Person's r correlations for *Kat1*, *Kat3* and *Got2/Kat4* in wild-type skeletal muscle were calculated (>0,4, $p < 0,05$) followed by DAVID functional clustering analysis[51] in each independent network and in overlapping networks. To assess the KATs-associated network in human skeletal muscle we performed Kats-targeted approach of overlapping networks using a tissue-integrated genome-scale analysis (archived in http://giant.princeton.edu)[25]. This was followed by DAVID functional clustering.

To evaluate interacting protein-networks for the functional units of the malate-aspartate shuttle we used data from soluble protein-protein-interaction complexes evaluated across diverse metazoan models (data stored in http://metazoa.med.utoronto.ca)

**Statistical analysis.** Data is expressed as average + SEM. For in vitro experiments, $n$ indicates independent experiments with at least three technical replicates. The number of biological replicates used was based on our previous experience with the different protocols used. For in vivo experiments, $n$ indicates number of mice per group. The number of animals used in each experiment was counted accordingly with our previous experience and standards within the field. This is the minimal number of mice needed to achieve statistical significance for most of the tests performed. All subjects (control and test) were randomized according to age, gender and weight. All statistical analyses (including outliers test) were performed using GraphPad Prism 6. Unpaired student's $t$-test was used when two groups were compared, and one-way analysis of variance (ANOVA) followed by Fisher's least significance difference (LSD) test for post hoc comparisons was used to compare multiple groups. Statistical significance was defined as $p < 0.05$. Correlations were calculated by Pearson correlation.

**Reporting summary.** Further information on research design is available in the Nature Research Reporting Summary linked to this article.

## Data availability

Authors can confirm that all relevant data are included in the paper and/or its supplementary information files.

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

## Acknowledgements

We would like to thank all members of the lab for insightful discussions about the project. This work was supported by grants from the Strategic Research Programme in Diabetes at Karolinska Institutet (J.L.R., J.T.L.), the Novo Nordisk Foundation (Denmark) (J.L.R.), the Swedish Diabetes Foundation (J.L.R) and the Swedish Research Council (J.L.R., J.T.L.), the Swiss National Science Foundation (C.H.), and the European Research Council (616830-MUSCLE_NET to C.H.). L.Z. is supported by the Chinese Scholar Council, L.K. is supported by a Novo Nordisk postdoctoral fellowship run in partnership with Karolinska Institutet, and D.M.S.F. was supported in part by a postdoctoral fellowship from the Wenner-Gren Foundations. We thank Dr. Jorgina Satrústegui (Universidad Autonoma de Madrid) for the kind gift of the Slc25a12 antibody.

## Author contributions

L.Z.A., D.M.S.F. and J.L.R. conceived, coordinated and designed the study. L.Z.A., D.M.S.F., and S.D. performed and analyzed animal experiments, tissue culture, in vitro experiments and gene expression with contributions from I.C., L.K., M.I. and R.F. L.Z.A. performed computational analyses. D.M.S.F. and L.Z.A. performed and analyzed seahorse respirometry experiments. L.Z and J.T.L. performed skeletal muscle fatigue measurements. R.F. and C.H. performed and analyzed the muscle-specific PGC-1α KO exercise experiments and corresponding gene expression data. T.V., M.B. and S.K. performed exercise interventions in humans. L.Z.A., D.M.S.F., S.D. and J.L.R. wrote the manuscript. All authors reviewed the results and approved the final version of the manuscript.

**Additional information**

**Competing interests:** The authors declare no competing interests.

