## [Peer Review File · Nature Communications]

Reviewers' Comments:

Reviewer #1:

Remarks to the Author:

The main and novel finding of the manuscript supports enhanced expression of both glycolysis and MAS genes with kynurenine metabolic use to increase bioenergetic efficiency during exercise training in skeletal muscle. The activation of MAS supported by kynurenine catabolism is here proposed as part of the physiological mechanism implicated in adaptation to endurance exercise and fatigue-resistance. This work shows quite convincing and very relevant data supporting the physiological role of MAS activation and Kyn utilization in fatigue-resistance after endurance exercise. However, some critical points would need further clarification :

1- Herein, Aminoxyacetate (AOA) is used to inhibit the NADH malate-aspartate shuttle (MAS) activity (figure 3). AOA is moderately potent as an inhibitor of aspartate aminotransferase and therefore concentrations of 1-5 mM are needed. However, at these doses several other aminotransferases are likely to be inhibited and for this reason this is not more used as a specific tool for MAS inhibition. Besides, the role of MAS in the kynurenine use and effects is a novel finding herein reported that might be extensively proved. So, a genetic disruption of MAS activity (i.e., silencing *Slc25a12*) would be much more accurate to show the direct impact of MAS inhibition on functional contribution in myotubes.

In this condition (silencing *Slc25a12*), OCR and aspartate content might be assayed with and without kynurenine treatment. In normal conditions aspartate synthesis occurs in mitochondria and is dependent on MAS activity; however, in some anaerobic or other situations, cytosolic GOT1 might reverse to generate aspartate (in a MAS-independent way).

2- In skeletal muscle, PGC-1 α 1 induces the expression of glycolysis and MAS genes; in this way glycolysis-derived pyruvate might be elevated and "pulled" through the mitochondrial pyruvate carrier into mitochondria under these conditions. In order to further investigate the role of PGC-1 α 1 and kynurenine metabolism in the functional coupling between glycolytic and mitochondrial metabolism, it would be useful to evaluate PDH activity as activation of PDH complex will pull pyruvate from the cytosol to mitochondria to increase mitochondrial OCR in excitable cells. For this purpose, it would be clarifying to measure the ratio between PDH and P-PDH in skeletal muscle from Mck-PGC-1 α 1 mice with and without kynurenine treatment. Although *pdha* seems to be highly upregulated in skeletal muscle of Mck-PGC-1 α 1 mice with kynurenine treatment (Fig S6B). Also, protein quantification for AGC (*Slc25a12*) and OGC (*Slc25a11*) transporters is lacking in these Mck-PGC-1 α 1 mice with and without kynurenine treatment.

3- There must be some mistake with the label for the oxoglutarate-malate carrier (OGC). In Figure 1K, OGC is designated as *Slc25a10* instead of *Slc25a11*. However, throughout the manuscript when referring to MAS components the same nomenclature (*Slc25a10*) is used (Figure 2B, 2E, 4E, 4G, Fig. S4, S5A). At this point, something is wrong or only misspelled.

4- In the manuscript, it is explained that skeletal muscle from mko-PGC-1 α mice do not have the ability to use kynurenine to support bioenergetics; indeed, they have reduced levels of genes involved in energy metabolism under baseline and kynurenine administration (Fig 2D-2E and fig. S6C-S6D). However, it is interesting to observe a highly significant increment in the transcripts for *Ldha* with kynurenine treatment (S6D). This point is important to be clarified as it might be showing an effect of kynurenine on lactate utilization or oxidation in the intracellular lactate shuttle. This lactate shuttle, other than MAS or glycerol-3-phosphate shuttle (G3PS) might be participating to bioenergetics fueling in skeletal muscle (Brooks GA, review Cell Metab 2018). Also it would be useful to see data for G3PS in these mko-PGC-1 α mice

MINOR points.

1- The title of the Figure S7 is misspelled

2- Why the SeaHorse assays have been performed under 25 mM glucose conditions? As far as I know, these hyperglycemic conditions (those of the cell culture) could make more difficult to observe the effects of different conditions, substrates or stimuli.

Reviewer #2:

Remarks to the Author:

Agudelo and colleagues studied how the transcriptional coregulator PGC1alpha affects the malate-aspartate shuttle in skeletal muscle. They demonstrated a link between kynurenine metabolism, glutamate and aspartate biosynthesis, and hence metabolite accumulation, in this context. While several lines of evidence confirm this hypothesis, some gaps and questions remain.

1.) The authors should emphasize the conceptual advances of their study. First, due to the strong effect of PGC1alpha on mitochondrial biogenesis and function, and increase in gene expression, most of which are mitochondrial, might be expected. While I readily acknowledge that "expected" is not show, second, more importantly, the control of the malate aspartate shuttle as well as aspartate and glutamate levels has previously been reported (Hatazawa et al. PLOS ONE 2015, PMID 26114427 – a study which probably should be cited here).

2.) The authors link their postulated mechanism to an increase in glycolysis. However, most prior data revealed that PGC1alpha strongly reduces glycolysis in muscle, e.g. as demonstrated by Wende et al. JBC 2007, PMID 17932032, who have demonstrated reduced glycolytic flux and expression of phosphofructokinase in PGC1alpha inducible muscle-specific transgenic mice. Downstream, PGC1alpha-controlled elevation of PDK4 would further throttle glucose oxidation (Wende et al. Mol Cell Biol 2005, PMID 16314495). Thus collectively, the PGC1alpha-mediated increase in glucose uptake leads to a rerouting of glucose to glycogen and the pentose phosphate pathway, but not glycolysis. How do the authors reconcile their hypothesis with these observations?

3.) It is somewhat surprising that Kyn administration to control mice fails to have an effect (e.g. see Fig. 2 and S6), even though endogenous PGC1alpha expression in skeletal muscle is considerable even in the absence of transgenic overexpression. How do the authors explain this observation? It might be interesting to exercise Kyn-treated control mice, which then could be expected to exhibit an effect? Along the same lines, absence of a positive effect of Kyn treatment in PGC1alpha knockouts is expected if PGC1alpha were necessary for this. The authors however report a negative effect. What is the hypothesis for this?

4.) It is surprising that in many of the experiments, gain- and loss-of-function of PGC1alpha fails to significantly affect the expression of many genes that have previously been reported to be altered in the same in vitro and in vivo experimental systems, for example, most of the transcription factors in Fig. 2 and S5, and many of the mitochondrial genes shown in Fig. S5 and S6 (shown in numerous papers). In fact, some of these genes appear to be significantly elevated in the transcriptomics analysis (Supplemental Table 1)?

5.) To judge the results of the training study, the authors should indicate the time of sacrifice of the mice after the last bout of running wheel activity to discern acute from chronic effects. Moreover, some measure of the running activity of these mice would be helpful to assess the training.

Reviewer #3:

Remarks to the Author:

Agudelo et al. examined the role of PGC-1 alpha in regulating kynurenine metabolism in skeletal muscle. Using a combination of muscle specific PGC-1 alpha TG mice and cultured muscle cells the authors provide strong evidence that PGC-1 alpha activates the malate-aspartate shuttle (MAS) in skeletal muscle and that this is supported by kynurenine catabolism. The data presented in this paper are novel and should be of interest to those investigating skeletal muscle metabolism. The authors conclude that these alterations could be "...part of the adaptations to endurance exercise". Exercise clearly induces PGC-1 alpha and increases the expression of genes involved in the MAS (current work). Moreover, the current data shows a clear effect of PGC-1 alpha on the expression of components of the MAS machinery. However, there is no evidence presented in the current manuscript that PGC-1 alpha is required for the effects of exercise. This needs to be addressed and could be done so through using PGC-1 alpha knockout mice. Related to this point, the authors need to provide data showing the relative increases in PGC-1 alpha expression with overexpression compared to what is seen following exercise training in mice and humans. This is an important point as it would shed insight into the physiological relevance of their over-expression models. In addition to the major points highlighted above there are methodological/experimental points that need to be addressed as follows:

1. Throughout the manuscript gene expression data is presented as fold increases compared to a control condition which is shown as a dotted line. This does not provide any insight into the degree of variability in the control conditions. The gene expression data needs to be presented similar to how the aspartate and malate data have been shown.
2. The authors provide data in which KAT1,3 and 4 are knocked down in muscle cells via siRNA and then OCR measured. The degree of knockdown needs to be presented.
3. Throughout the manuscript "N" values for the cell culture, animal experiments and human training intervention are not given. These need to be provided.
4. How long after the last bout of exercise (both the human and rodent studies) were muscle samples harvested? Could the effects of training on the endpoints examined actually be due to the residual effects of the last bout of exercise? Please address.
5. For the long-term training studies the gene expression data needs to be confirmed by measuring the protein content of constituents of the MAS.
- 6t. It is not clear from the methods what skeletal muscle was harvested from mice. Please provide this information in the methods and specify in the manuscript. Could there be fiber type specific effects of PGC-1 alpha and/or exercise on the MAS?

Reviewer #4:

Remarks to the Author:

To the authors

This paper reports quite a comprehensive and pretty compelling data-set linking kynurenine metabolism in skeletal muscle to energy metabolism and fatigue resistance in a PGC1a dependent manner. A series of complimentary studies are presented that jointly nicely build the story. Most of the findings are novel and provide new insights into the complexity of skeletal muscle energy metabolism and the interaction of oxidative and non-oxidative and cytosolic vs mitochondrial metabolism. Most of the experiments appear to be conducted carefully with state-of-the-art data analysis. However, there are a few predominantly minor concerns that deserve attention during revision.

The paper nicely shows that the increased capacity to convert kynurenine to kynurenic acid in the muscle upon overexpression of PGC1a promotes malate-aspartate dependent shuttling of glycolysis derived NADH into the mitochondria and anapleurotic feeding of the TCA cycle with Got2-mediated Glutamate. This indeed may provide the biochemical basis for increased generation of ATP from the same amount of substrate input, one could call this increased energy efficiency. However, changes in ATP availability have not been reported. Alternatively, one can talk about increased energy efficiency is more work is being produced with the same energy input, but

external work has not been measured in the present study either. Hence, I think the current title is an overstatement of what has been observed and deserved removal of the 'energy efficiency' part.

Anapleurotically feeding the TCA cycle with additional carbons may indeed result in production of more NADH/FADH₂ and hence ATP synthesis. If I understood the concept of the paper correctly, it is glutamate that in this model feeds into the TCA cycle. Most likely glutamate derived carbons feed into the cycle as α-ketoglutarate. α-ketoglutarate, however, also is the substrate to convert kynurenine into kynurenic acid whilst releasing glutamate. If this is correct (which I must admit I am not sure of), this would imply that there is no net increase of carbons feeding the TCA cycle. The authors are invited to reflect upon this concern in a rebuttal and if my reasoning is correct, to discuss the anapleurotic feeding into this perspective.

In their cell model, the authors show that elevating PGC 1α suffices to increase glutamate level and induces genes involved in malate aspartate shuttling that is further augmented by kynurenine supplementation to the cells, but only upon overexpression of PGC 1α. This appears to be a dose dependent phenomenon, both with respect to PGC 1α levels (wt muscles do not show this) as well as for Kynurenine concentrations, as 'high' concentrations of Kynurenine abolishes mitochondrial respiration and impedes aspartate biosynthesis. From the mechanisms suggested, it is not straightforward to explain this dual dose dependency. The authors are encouraged to discuss this dual dose dependency in the perspective of the proposed mechanisms, particularly as it may impact on the clinical application of this study and the window of dosing of kynurenine, if exogenously supplemented (most humans do not over-express PGC 1α...).

It is claimed that most of the effects observed require functional PGC 1α. Most of the data presented indeed are in line with this statement. To firmly underpin the statement, however, PGC 1α silencing or knock-out is the model of choice. Although the authors use these models, the only report very little information on these experiments nor on the models used (if I read correctly, the paper for example does not provide information on how PGC 1α was downregulated in the primaries and whether PGC 1α was completely absent or silenced and hence still may have residual activity. This is of particular interest in face of the apparent effects of the level of PGC 1α on kynurenine metabolism. Please elaborate more on your knock-down/knock-out data or make the statements on requirement of (high levels?) of PGC 1α less firm.

In a previous paper (Hoeks, J. et al. J Cell Physiol 2011), it has been shown that I mitochondrial isolated from the same straining of PGC 1α overexpressing mice, possess elevated respiration on lipid derived substrates and not on glycolytic substrates (this even seems to be a bit compromised). These are mitochondria studied in isolation, so one can argue that cytosolic processes are not been taken into account, nevertheless the buffers for respiration contain high levels of glutamate and malate. The paper could gain some 'balance' by also mentioning these observations, which at first glance can not be readily reconciled with the current study.

The data on running capacity (with a non-significant drop in running time upon carbidopa supplementation) are in my opinion not very convincing. These studies have been performed in WT mice, probably with normal/low PGC 1α level and hence little reason to anticipate increased KAT activity. So, one can wonder if this is the best model to examine the effects of KAT inhibition by carbidopa. I would suggest omitting these data (and the discussion related to it) from the manuscript or to add data on PGC 1α levels and KAT activity with and without carbidopa and correlate these activities to running performance to substantiate the findings.

The final sentence of the paper 'In addition, the observation that carbidopa inhibits...has important clinical observation', must be removed from the paper. There is no indication from the paper that underpins the suggestion of clinical relevance, moreover, if I recall correctly, carbidopa is seldomly being prescribed for a single drug treatment, always in combination with levodopa and these combined effects have not been examined in the present study.

POINT-BY-POINT RESPONSE TO REVIEWERS

We would like to thank the reviewers for their time and comments on the manuscript. We were happy to see that the reviewers appreciated our work and we have tried to address all of the editor's and the reviewers' comments and suggestions. This was done by adding new data, clarifying the text, or both. We feel that the suggestions have really improved the manuscript. Below you can find a point-by-point response to the questions that were raised.

Reviewer #1 (Remarks to the Author):

The main and novel finding of the manuscript supports enhanced expression of both glycolysis and MAS genes with kynurenine metabolic use to increase bioenergetic efficiency during exercise training in skeletal muscle. The activation of MAS supported by kynurenine catabolism is here proposed as part of the physiological mechanism implicated in adaptation to endurance exercise and fatigue-resistance. This work show quite convincing and very relevant data supporting the physiological role of MAS activation and Kyn utilization in fatigue-resistance after endurance exercise. However, some critical points would need further clarification:

1- Herein, Aminooxyacetate (AOA) is used to inhibit the NADH malate-aspartate shuttle (MAS) activity (figure 3). AOA is moderately potent as an inhibitor of aspartate aminotransferase and therefore concentrations of 1-5 mM are needed. However, at these doses several other aminotransferases are likely to be inhibited and for this reason this is not more used as an specific tool for MAS inhibition. Besides, the role of MAS in the kynurenine use and effects is a novel finding herein reported that might be extensively proved. So, a genetic disruption of MAS activity (i.e., silencing Slc25a12) would be much more accurate to show the direct impact of MAS inhibition on functional contribution in myotubes.

In this condition (silencing Slc25a12), OCR and aspartate content might be assayed with and without kynurenine treatment. In normal conditions aspartate synthesis occurs in mitochondria and is dependent on MAS activity; however, in some anaerobic or other situations, cytosolic GOT1 might reverse to generate aspartate (in a MAS-independent way).

The reviewer raises an important point. We used AOA at 0.1 mM and this is now specified in the figure legend and methods. Under these conditions we observed a reduction in basal and maximal OCR (Fig. 3g) without changes in ECAR (data now added in Fig. S7e). However, we have performed the experiments that the reviewer requested, and measured OCR and aspartate levels following silencing of Slc25a12 in myotubes. We have also performed the same experiments upon Slc25a11 silencing. These data are included in the manuscript (Fig. S7g-S7i). By using transient siRNA transfections, we reduced Slc25a12 and Slc25a11 expression by 44% and 34%, respectively (Fig. S7f). Under these conditions we did not observe any effects on OCR, even when Kyn was added (Fig. S7g). Although we didn't achieve complete ablation of transporter expression, these results could indeed support the reviewer's comment that the AOA treatment (even at this lower concentration) could have off-target effects. Interestingly, Slc25a12 silencing elevated ECAR values at baseline with little effect of Kyn addition, whereas Slc25a11 silencing had a more robust effect on ECAR values upon Kyn addition. In a similar fashion, Slc25a12 knockdown resulted in significantly elevated aspartate levels that were not affected by Kyn, whereas lowering Slc25a11 expression had no effect (despite a trend towards elevated

aspartate after Kyn). This difference between the chemical inhibition and the silencing of the transporters is now indicated in the text.

In any case, if the reviewer finds it preferable, we can remove the AOA data and include only the new silencing data, although in our view they could be complementary since we are inhibiting the MAS at several different points.

2- In skeletal muscle, PGC-1 α 1 induces the expression of glycolysis and MAS genes; in this way glycolysis-derived pyruvate might be elevated and “pulled” through the mitochondrial pyruvate carrier into mitochondria under these conditions. In order to further investigate the role of PGC-1 α 1 and kynurenine metabolism in the functional coupling between glycolytic and mitochondrial metabolism, it would be useful to evaluate PDH activity as activation of PDH complex will pull pyruvate from the cytosol to mitochondria to increase mitochondrial OCR in excitable cells. For this purpose, it would be clarifying to measure the ratio between PDH and P-PDH in skeletal muscle from Mck-PGC-1 α 1 mice with and without kynurenine treatment. Although pdha seems to be highly upregulated in skeletal muscle of Mck-PGC-1 α 1 mice with kynurenine treatment (Fig S6B).

We have determined the levels of PDH and P-PDH, which are now shown in Fig. S6c. After quantification of the western blot results, there is a trend towards an increase in the P-PDH/PDH ratio upon Kyn treatment (1.37-fold), but that does not reach statistical significance (p=0.139). This information is now included in the manuscript.

Also, protein quantification for AGC (Slc25a12) and OGC (Slc25a11) transporters is lacking in these Mck-PGC-1 α 1 mice with and without kynurenine treatment.

This is now shown in Fig. 2c. We observed a 5.4- and 7-fold increase in AGC in Mck-PGC-1 α 1 treated with PBS and Kyn, respectively, and vs wt PBS. OGC protein levels were barely detectable in wt muscle (Fig 2c), and no apparent changes were observed upon Kyn administration to wt or Mck-PGC-1 α 1 mice.

3- There must be some mistake with the label for the oxoglutarate-malate carrier (OGC). In Figure 1K, OGC is designated as Slc25a10 instead of Slc25a11. However, throughout the manuscript when referring to MAS components the same nomenclature (Slc25a10) is used (Figure 2B, 2E, 4E, 4G, Fig. S4, S5A). At this point, something is wrong or only misspelled.

The reviewer is correct and we apologize for the spelling mistake. It is now corrected throughout the manuscript.

4- In the manuscript, it is explained that skeletal muscle from mko-PGC-1 α mice do not have the ability to use kynurenine to support bioenergetics; indeed, they have reduced levels of genes involved in energy metabolism under baseline and kynurenine administration (Fig 2D-2E and fig. S6C-S6D). However, it is interesting to observe a highly significant increment in the transcripts for Ldha with kynurenine treatment (S6D). This point is important to be clarified as it might be showing an effect of kynurenine on lactate utilization or oxidation in the intracellular lactate shuttle. This lactate shuttle, other than MAS or glycerol-3-phosphate shuttle (G3PS) might be participating to bioenergetics fueling in skeletal muscle (Brooks GA, review Cell Metab 2018).

We carefully reviewed our data in light of the reviewer’s comment, and the indicated review paper. In this context, an activation of the intracellular lactate shuttle should result in an increase in oxygen consumption. However, what we

see is precisely the opposite, a reduction in OCR and an increase in ECAR (indicative of lactate secretion). In agreement, it has been previously shown that MKO-PGC-1 α mice have increased circulating lactate levels (Liu J. et al., *EMBO Mol Med.* 2016).

Also it would be useful to see data for G3PS in these mko-PGC-1 α mice.

We now show the expression of G3PS genes in MKO-PGC-1 α 1 mice in Fig. S7b.

MINOR points.

1- The title of the Figure S7 is misspelled

That is now corrected in the manuscript.

2- Why the SeaHorse assays have been performed under 25 mM glucose conditions? As far as I know, these hyperglycemic conditions (those of the cell culture) could make more difficult to observe the effects of different conditions, substrates or stimuli.

We agree that the high glucose could be closing the observation window, but since these are the standard conditions that we use for primary myotube culture we didn't want to stress the cells in any other way than the treatment being tested.

Reviewer #2 (Remarks to the Author):

Agudelo and colleagues studied how the transcriptional coregulator PGC1alpha affects the malate-aspartate shuttle in skeletal muscle. They demonstrated a link between kynurenine metabolism, glutamate and aspartate biosynthesis, and hence metabolite accumulation, in this context. While several lines of evidence confirm this hypothesis, some gaps and questions remain.

1.) The authors should emphasize the conceptual advances of their study. First, due to the strong effect of PGC1alpha on mitochondrial biogenesis and function, and increase in gene expression, most of which are mitochondrial, might be expected. While I readily acknowledge that "expected" is not show, second, more importantly, the control of the malate aspartate shuttle as well as aspartate and glutamate levels has previously been reported (Hatazawa et al. PLOS ONE 2015, PMID 26114427 – a study which probably should be cited here).

Indeed, one of the data sets that we used for our initial multi-omics analysis was the one reported by Hatazawa et al., in PLOS One 2015. This is stated right at the beginning of the manuscript and the article is cited there and then again in methods under "transcriptomic and metabolomic analysis". We have now also mentioned it in a new paragraph of concluding remarks.

Our message is not only that PGC-1 α 1 activates the MAS in muscle, showing here the gene networks involved, and data from *in vitro* and *in vivo* manipulation of the system. It is also that in the trained state (i.e. high PGC-1 α 1 levels), skeletal muscle can use Kyn (a compound with known neurotoxic properties) to support energy metabolism. We also show that in the absence or at low levels of PGC-1 α , Kyn (or its metabolites) has mostly toxic effects, even inhibiting

mitochondrial respiration. We have highlighted this information throughout the manuscript.

2.) *The authors link their postulated mechanism to an increase in glycolysis. However, most prior data revealed that PGC1alpha strongly reduces glycolysis in muscle, e.g. as demonstrated by Wende et al. JBC 2007, PMID 17932032, who have demonstrated reduced glycolytic flux and expression of phosphofructokinase in PGC1alpha inducible muscle-specific transgenic mice. Downstream, PGC1alpha-controlled elevation of PDK4 would further throttle glucose oxidation (Wende et al. Mol Cell Biol 2005, PMID 16314495). Thus collectively, the PGC1alpha-mediated increase in glucose uptake leads to a rerouting of glucose to glycogen and the pentose phosphate pathway, but not glycolysis. How do the authors reconcile their hypothesis with these observations?*

The reviewer is correct in pointing out that different mouse models of elevated PGC-1 α 1 expression in skeletal muscle have not always generated overlapping results (Mck-PGC-1 α 1, Lin J. et al. Nature 2002; PGC-1 α TRE (+), Wende et al. JBC 2007; HSA-PGC-1 α -b, Miura et al., Endocrinology 2008). The 3 models use different strategies to elevate PGC-1 α in muscle. Mck- and HSA-driven PGC-1 α expression is already activated at birth and remains on during the life of the mice, whereas the model referred to by the reviewer is an inducible model (mice are kept on doxycycline, which represses transgene expression until it is removed from the diet). Evidence of differences between the models is reflected on exercise performance, Glut4 levels, glucose tolerance, insulin sensitivity, among other. Namely, the mouse model that the reviewer refers to (PGC-1 α TRE (+)) has been shown to have a reduction in exercise capacity (Wende et al., JBC 2017), whereas the Mck-PGC-1 α mouse has increased exercise performance (Calvo et al., JAP 2008). Interestingly, both models show increased muscle glycogen content at rest but the Mck-PGC-1 α can use it during exercise (Kiilerich et al., AJP-RICP, 2010) whereas the PGC-1 α TRE (+) cannot (Wende et al., JBC 2017). As it is very difficult to compare the levels of PGC-1 α overexpression between the papers/models, we can only say that they fundamentally differ on chronic vs acute PGC-1 α elevation.

In addition, the effect of PGC-1 α on PDK4 referred to by the reviewer, which was observed in C2C12 cells, was also observed in the Mck-PGC-1 α 1 mice, but not always correlated with increased P-PDH levels (Kiilerich et al., AJP-RICP, 2010).

These aspects are now discussed in the text.

3.) *It is somewhat surprising that Kyn administration to control mice fails to have an effect (e.g. see Fig. 2 and S6), even though endogenous PGC1alpha expression in skeletal muscle is considerable even in the absence of transgenic overexpression. How do the authors explain this observation? It might be interesting to exercise Kyn-treated control mice, which then could be expected to exhibit an effect? Along the same lines, absence of a positive effect of Kyn treatment in PGC1alpha knockouts is expected if PGC1alpha were necessary for this. The authors however report a negative effect. What is the hypothesis for this?*

Since muscle Kat expression is dependent on PGC-1 α 1 (Agudelo et al., Cell 2014), MKO and wt myotubes will have a reduced capacity to convert Kyn into Kyna, thus allowing its degradation through the kynurenine pathway. Interestingly, KP metabolites such as 3-hydroxykynurenine and 3-hydroxyanthranilic acid have been shown to impair energy metabolism by direct

inhibition of mitochondrial respiration and ROS production Nagamure et al. Adv. Exp. Med. Biol. 1999; Okuda et al. J. Neurochem. 1998).

4.) *It is surprising that in many of the experiments, gain- and loss-of-function of PGC1alpha fails to significantly affect the expression of many genes that have previously been reported to be altered in the same in vitro and in vivo experimental systems, for example, most of the transcription factors in Fig. 2 and S5, and many of the mitochondrial genes shown in Fig. S5 and S6 (shown in numerous papers). In fact, some of these genes appear to be significantly elevated in the transcriptomics analysis (Supplemental Table 1)?*

We thank the reviewer for his/her point. We wonder if the y-axis might be misleading so if needed, we can replot the graphs. We can observe an increase in Esrra, PPAR α NRF1, NRF2 almost 2-fold and 1.5-fold for MEF2 in both Mck-PGC-1 α 1 and myotubes transduced with PGC-1 α 1 adenovirus in accordance with what was described before (e.g. Lin et al. Nature 2002; Ruas et al. Cell 2012; Agudelo et al. Cell 2014). In a similar way, we also observe a decrease in the expression of several transcription factors in MKO-PGC-1 α mice (0.5- to 0.8-fold), similar to what was reported before (Handschin et al. J Biol Chem 2007; Handschin et al. J Clin Invest 2007).

5.) *To judge the results of the training study, the authors should indicate the time of sacrifice of the mice after the last bout of running wheel activity to discern acute from chronic effects. Moreover, some measure of the running activity of these mice would be helpful to assess the training.*

We thank the reviewer for raising this important point. We have clarified in our methods the time of sacrifice of the mice (12 h after the last bout of exercise). In addition, we enclose the running activity of the mice when performing voluntary exercise. We can add this information to the manuscript if necessary.

Reviewer #3 (Remarks to the Author):

Agudelo et al. examined the role of PGC-1 alpha in regulating kynurenine metabolism in skeletal muscle. Using a combination of muscle specific PGC-1 alpha TG mice and cultured muscle cells the authors provide strong evidence that PGC-1 alpha activates the malate-aspartate shuttle (MAS) in skeletal muscle and that this is supported by kynurenine catabolism. The data presented in this paper are novel and should be of interest to those investigating skeletal muscle metabolism. The authors conclude that these alterations could be “..part of the adaptations to endurance exercise”. Exercise clearly induces PGC-1 alpha and increases the expression of genes involved in the MAS (current work). Moreover, the current data shows a clear effect of PGC-1 alpha on the expression of components of the MAS machinery. However, there is no evidence presented in the current manuscript that PGC-1 alpha is required for the effects of exercise. This needs to be addressed and could be done so through using PGC-1 alpha knockout mice.

The effects of muscle PGC-1 α 1 expression on promoting adaptations to exercise have been previously reported by using either the same transgenic mice we use here (Mck-PGC-1 α mice) or muscle KOs (MKO-PGC-1 α), as suggested by the reviewer. Indeed, Mck-PGC-1 α 1 mice have been shown to have higher exercise capacity (Calvo et al., JAP 2008), which has been correlated to increased mitochondrial content and a fiber-type switch towards more oxidative fibers (Lin J et al., Nature 2002) and increased angiogenesis (Arany et al., Nature 2008). Conversely, muscle-specific deletion of the PGC-1 α gene results in decreased exercise performance and marks of inflammation in muscle (Handschin et al. J Biol Chem 2007; Handschin et al. J Clin Invest 2007), which aggravate with age (Sczelecki et al., AJP- EM 2014). Human data is also available although correlative (Mathai J Appl Physiol 2008). This is now clarified in the introduction and these citations updated.

Related to this point, the authors need to provide data showing the relative increases in PGC-1 alpha expression with overexpression compared to what is seen following exercise training in mice and humans. This is an important point as it would shed insight into the physiological relevance of their over-expression models. In addition to the major points highlighted above there are methodological/experimental points that need to be addressed as follows:

This information has also been previously published. We have now clarified this in the text. In Mck-PGC-1 α 1 mouse the levels of PGC-1 α in glycolytic muscles are brought up to the levels seen in wild-type soleus (Lin et al. Nature 2002). This is translated into an increase of about 10-fold in muscles such as the EDL and the plantaris, vs the soleus where there is no further increase (Lin et al., Nature 2002). In human studies the increase in muscle PGC-1 α induced by exercise also varies within that range, depending on the muscle analysed and the detection methods used (Baar K et al., FASEB J 2002; Mathai J Appl Physiol 2008).

1. Throughout the manuscript gene expression data is presented as fold increases compared to a control condition which is shown as a dotted line. This does not provide any insight into the degree of variability in the control conditions. The gene expression data needs to be presented similar to how the aspartate and malate data have been shown.

This has been changed in all the figures and now the controls can be seen as a graph bar with its specific error bar depicting variability.

2. *The authors provide data in which KAT1,3 and 4 are knocked down in muscle cells via siRNA and then OCR measured. The degree of knockdown needs to be presented.*

This is now present in Fig. S7d where we show a 60 - 70 % decrease in KAT expression.

3. *Throughout the manuscript “N” values for the cell culture, animal experiments and human training intervention are not given. These need to be provided.*

We apologize for the missing information. This has been corrected in the manuscript.

4. *How long after the last bout of exercise (both the human and rodent studies) were muscle samples harvested? Could the effects of training on the endpoints examined actually be due to the residual effects of the last bout of exercise? Please address.*

This is indeed an important point. To avoid confounding acute effects from the exercise intervention, muscle samples were collected 12 hours after the last bout of exercise in both studies. This information is now included in the methods section.

5. *For the long-term training studies the gene expression data needs to be confirmed by measuring the protein content of constituents of the MAS.*

We understand the reviewer’s comment but unfortunately we don’t have any more muscle biopsies left from the long-term training studies, which would be difficult to repeat in a timely fashion (especially from the human study). However, we have looked at publicly available data on muscle proteomics from mouse or human long-term training where we could see an increase in several proteins of the MAS. This information is now included in Fig. 5c (murine exercise) and Fig. 5f (human exercise) and it is discussed in the manuscript.

6t. *It is not clear from the methods what skeletal muscle was harvested from mice. Please provide this information in the methods and specify in the manuscript. Could there be fiber type specific effects of PGC-1 alpha and/or exercise on the MAS?*

This information has now been added to the methods section of the manuscript. For the mouse studies, we collected the gastrocnemius muscle since it is a mixed muscle. Indeed, Mck-PGC-1 α 1 transgenic mice have been reported to have a switch towards slow fibers (Lin et al. Nature 2002). Interestingly, Schantz and Henriksson (Acta Physiol Scand. 1987) have shown that type I skeletal muscle fibers (which have the highest levels of PGC-1 α 1 when compared to type II fibers) exhibit higher levels of the MAS enzymes. Also, in this paper, the authors show that after endurance training (modeled here by the Mck-PGC-1 α 1 transgenic mice) both fiber types have higher levels of MAS enzymes. This is now also discussed in the text.

Reviewer #4 (Remarks to the Author):

To the authors

This paper reports quite a comprehensive and pretty compelling data-set linking kynurenine metabolism in skeletal muscle to energy metabolism and fatigue resistance in a PGC1a dependent manner. A series of complimentary studies are presented that jointly nicely build the story. Most of the findings are novel and provide new insights into the complexity of skeletal muscle energy metabolism and the interaction of

oxidative and non-oxidative and cytosolic vs mitochondrial metabolism. Most of the experiments appear to be conducted carefully with state-of-the-art data analysis. However, there are a few predominantly minor concerns that deserve attention during revision.

The paper nicely shows that the increased capacity to convert kynurenine to kynurenic acid in the muscle upon overexpression of PGC1a promotes malate-aspartate dependent shuttling of glycolysis derived NADH into the mitochondria and anapleurotic feeding of the TCA cycle with Got2-mediated Glutamate. This indeed may provide the biochemical basis for increased generation of ATP from the same amount of substrate input, one could call this increased energy efficiency. However, changes in ATP availability have not been reported. Alternatively, one can talk about increased energy efficiency is more work is being produced with the same energy input, but external work has not been measured in the present study either. Hence, I think the current title is an overstatement of what has been observed and deserved removal of the 'energy efficiency' part.

We have followed the reviewer's recommendation and complemented our results with ATP measurements. Indeed, ATP levels in myotubes transduced with PGC-1 α 1 adenovirus were higher than controls, and were further elevated upon Kyn supplementation (Fig. 3a). These data support our statement that PGC-1 α 1 reroutes Kyn metabolism to increase energy efficiency.

Anapleurotically feeding the TCA cycle with additional carbons may indeed result in production of more NADH/FADH₂ and hence ATP synthesis. If I understood the concept of the paper correctly, it is glutamate that in this model feeds into the TCA cycle. Most likely glutamate derived carbons feed into the cycle as α -ketoglutarate. α -ketoglutarate, however, also is the substrate to convert kynurenine into kynurenic acid whilst releasing glutamate. If this is correct (which I must admit I am not sure of), this would imply that there is no net increase of carbons feeding the TCA cycle. The authors are invited to reflect upon this concern in a rebuttal and if my reasoning is correct, to discuss the anapleurotic feeding into this perspective.

After carefully reviewing our data and the manuscript text, we agree with the reviewer that indeed we cannot say if glutamate is being used to support the MAS or the TCA cycle. We do agree that for the same molecule of glutamate it's either one or the other. We have therefore removed from the text the reference to glutamate anaplerosis as well as the schematic representation previously on Fig. S1b.

Fiber type specificity?

This is now discussed in the manuscript, also in light of human data showing that type I fibers have higher MAS component levels, and that endurance training increases MAS in both type I and II fiber (Schantz and Henriksson, *Acta Physiol Scand.* 1987)

In their cell model, the authors show that elevating PGC1a suffices to increase glutamate level and induces genes involved in malate aspartate shuttling that is further augmented by kynurenine supplementation to the cells, but only upon overexpression of PGC1a. This appears to be a dose dependent phenomenon, both with respect to PGC1a levels (wt muscles do not show this) as well as for Kynurenine concentrations, as 'high' concentrations of Kynurenine abolishes mitochondrial respiration and impedes aspartate biosynthesis. From the mechanisms suggested, it is not straightforward to explain this dual dose dependency. The authors are encouraged to

discuss this dual dose dependency in the perspective of the proposed mechanisms, particularly as it may impact on the clinical application of this study and the window of dosing of kynurenine, if exogenously supplemented (most humans do not over-express PGC1a...).

The reviewer raises an important point, which was also brought up by reviewer #2. This is now addressed in the text as follows:

Since muscle Kat expression is largely dependent on PGC-1 α 1 (Agudelo et al., Cell 2014), MKO and wt myotubes will have a reduced capacity to convert Kyn into Kyna, thus allowing its degradation through the kynurenine pathway. Interestingly, KP metabolites such as 3-hydroxykynurenine and 3-hydroxyanthranilic acid have been shown to impair energy metabolism by direct inhibition of mitochondrial respiration and ROS production Nagamure et al. Adv. Exp. Med. Biol. 1999; Okuda et al. J. Neurochem 1998).

It is claimed that most of the effects observed require functional PGC1a. Most of the data presented indeed are in line with this statement. To firmly underpin the statement, however, PGC1a silencing or knock-out is the model of choice. Although the authors use these models, they only report very little information on these experiments nor on the models used (if I read correctly, the paper for example does not provide information on how PGC1a was downregulated in the primaries and whether PGC1a was completely absent or silenced and hence still may have residual activity. This is of particular interest in face of the apparent effects of the level of PGC1a on kynurenine metabolism. Please elaborate more on your knock-down/knock-out data or make the statements on requirement of (high levels?) of PGC1a less firm.

We thank the reviewer for pointing out the missing information, and have now addressed it in the manuscript. Myoblasts were isolated from muscle-specific PGC-1 α KO mouse muscle, and hence no further genetic manipulation was used.

In a previous paper (Hoeks J Cell Physiol 2011), it has been shown that mitochondria isolated from the same strain of PGC1a overexpressing mice, possess elevated respiration on lipid derived substrates and not on glycolytic substrates (this even seems to be a bit compromised). These are mitochondria studied in isolation, so one can argue that cytosolic processes are not been taken into account, nevertheless the buffers for respiration contain high levels of glutamate and malate. The paper could gain some 'balance' by also mentioning these observations, which at first glance can not be readily reconciled with the current study.

Indeed, as the reviewer points out, removing the cytosolic component will also remove Kat1, Got1, Mdh1, glycolysis-derived NADH, as well as any Kynurenine (which is not routinely added to the media). Interestingly, our new data added to figures S7 shows that inhibition of the MAS by reducing Slc25a11, Slc25a12, Kat1, or Kat4 expression increases ECAR values in cultured myotubes. This would indicate that there is an integration between both processes. This is now added and discussed in the text, in the context of the reference the reviewer suggests.

The data on running capacity (with a non-significant drop in running time upon carbidopa supplementation) are in my opinion not very convincing.

The indication that time running upon carbidopa administration is indeed different from controls ($p=0.0005$). We apologize for this lapse, and have now corrected in Fig. S8c.

These studies have been performed in WT mice, probably with normal/low PGC1a level and hence little reason to anticipate increased KAT activity. So, one can wonder if this is the best model to examine the effects of KAT inhibition by carbidopa. I would suggest omitting these data (and the discussion related to it) from the manuscript or to add data on PGC1a levels and KAT activity with and without carbidopa and correlate these activities to running performance to substantiate the findings.

The final sentence of the paper 'In addition, the observation that carbidopa inhibits...has important clinical observation', must be removed from the paper. There is no indication from the paper that underpins the suggestion of clinical relevance, moreover, if I recall correctly, carbidopa is seldomly being prescribed for a single drug treatment, always in combination with levodopa and these combined effects have not been examined in the present study.

To address this point according to both the reviewer's and the editor's recommendations, we have now changed the sentence to "... could have important clinical implications". The reviewer is correct that carbidopa is not administered individually, only in combination with levodopa. However, even in that form it works as an inhibitor of PLP-dependent aspartate aminotransferases (such as Kats and DOPA decarboxylase). We have included a reference that extensively discusses the side effects and toxicity related to carbidopa administration (Hinz M. et al., Clin. Pharmacol. 2014).

Reviewers' Comments:

Reviewer #1:

Remarks to the Author:

In the present version of the paper by Agudelo et al. the critical points suggested to be clarified have been rigorously addressed. This work shows in an extensive way that PGC-1 α 1 activates the MAS in skeletal muscle, supported by Kyn utilization, as part of the adaptations to endurance exercise.

In my opinion the paper is well executed, solidly argued and has enough relevance and novelty for its publication. However, it is rather intriguing that silencing Slc25a12 in myotubes induces a 3-fold increment in aspartate content (figure S7i) whereas silencing Got2/KAT4 induces a clear reduction in aspartate (Figure 3i). It has been previously reported by different groups that there is a dramatic reduction of aspartate in mouse C2C12 myoblast AGC1/ARALAR knocked-down (in which AGC1 levels were about 8-fold higher than AGC2/CITRIN levels; Alkan HF et al., Cell Metab 2018), in brain and neurons from AGC1/ARALAR knockout mice (Jalil MA et al., JBC 2005) and in human brain (Wibom R et al., N. Engl J Med 2009).

Another question to be further discussed is that aspartate levels herein measured by using Abcam commercial kit (Figure S7i) are much higher than those reported in mice neurons and brain (Jalil MA et al., JBC 2005). Is there any reference available about aspartate levels measured with other methods (amino acid analyzer or HPLC/GC) in primary myotubes or mice skeletal muscle?.

Minor verification: In Figure 1K, it is represented Got2/Kat3 instead of Got2/Kat4.

Reviewer #2:

Remarks to the Author:

Original Point 1.) The authors should emphasize the conceptual advances of their study. First, due to the strong effect of PGC1 α on mitochondrial biogenesis and function, and increase in gene expression, most of which are mitochondrial, might be expected. While I readily acknowledge that "expected" is not shown, second, more importantly, the control of the malate aspartate shuttle as well as aspartate and glutamate levels has previously been reported (Hatazawa et al. PLOS ONE 2015, PMID 26114427 – a study which probably should be cited here).

Authors reply: Indeed, one of the data sets that we used for our initial multi-omics analysis was the one reported by Hatazawa et al., in PLOS One 2015. This is stated right at the beginning of the manuscript and the article is cited there and then again in methods under "transcriptomic and metabolomic analysis". We have now also mentioned it in a new paragraph of concluding remarks. Our message is not only that PGC-1 α 1 activates the MAS in muscle, showing here the gene networks involved, and data from in vitro and in vivo manipulation of the system. It is also that in the trained state (i.e. high PGC-1 α 1 levels), skeletal muscle can use Kyn (a compound with known neurotoxic properties) to support energy metabolism. We also show that in the absence or at low levels of PGC-1 α , Kyn (or its metabolites) has mostly toxic effects, even inhibiting mitochondrial respiration. We have highlighted this information throughout the manuscript.

Reviewer rebuttal: There might have been a misunderstanding here: my intention was not to criticize an absence of citation of Hatazawa et al., but a lack of attribution of their findings in regard to PGC1 α regulation of the malate and aspartate shuttle as well as aspartate and glutamate levels, which also is a central aspect of the current manuscript. The authors have now partially rectified this by adding the sentence "The same was observed in the muscle metabolomics data reported for the HSA-PGC-1 α -b mice⁵", referring to the elevation of glutamate and aspartate. I fail to see reference 5 in the new paragraph of concluding remarks as stated by the authors in their reply. I agree that the other points regarding novelty are now more clearly highlighted in the text.

Original Point 2.) The authors link their postulated mechanism to an increase in glycolysis. However, most prior data revealed that PGC1 α strongly reduces glycolysis in muscle, e.g. as demonstrated by Wende et al. JBC 2007, PMID 17932032, who have demonstrated reduced

glycolytic flux and expression of phosphofructokinase in PGC1alpha inducible muscle-specific transgenic mice. Downstream, PGC1alpha-controlled elevation of PDK4 would further throttle glucose oxidation (Wende et al. Mol Cell Biol 2005, PMID 16314495). Thus collectively, the PGC1alpha-mediated increase in glucose uptake leads to a rerouting of glucose to glycogen and the pentose phosphate pathway, but not glycolysis. How do the authors reconcile their hypothesis with these observations?

Authors reply: The reviewer is correct in pointing out that different mouse models of elevated PGC-1a1 expression in skeletal muscle have not always generated overlapping results (Mck-PGC-1a1, Lin J. et al. Nature 2002; PGC-1a TRE (+), Wende et al. JBC 2007; HSA-PGC-1a-b, Miura et al., Endocrinology 2008). The 3 models use different strategies to elevate PGC-1a in muscle. Mck- and HSA-driven PGC-1a expression is already activated at birth and remains on during the life of the mice, whereas the model referred to by the reviewer is an inducible model (mice are kept on doxycycline, which represses transgene expression until it is removed from the diet). Evidence of differences between the models is reflected on exercise performance, Glut4 levels, glucose tolerance, insulin sensitivity, among other. Namely, the mouse model that the reviewer refers to (PGC-1a TRE (+)) has been shown to have a reduction in exercise capacity (Wende et al., JBC 2017), whereas the Mck-PGC-1a mouse has increased exercise performance (Calvo et al., JAP 2008). Interestingly, both models show increased muscle glycogen content at rest but the Mck-PGC-1a can use it during exercise (Kiilerich et al., AJP-RICP, 2010) whereas the PGC-1a TRE (+) cannot (Wende et al., JBC 2017). As it is very difficult to compare the levels of PGC-1a overexpression between the papers/models, we can only say that they fundamentally differ on chronic vs acute PGC-1a elevation. In addition, the effect of PGC-1a on PDK4 referred to by the reviewer, which was observed in C2C12 cells, was also observed in the Mck-PGC-1a1 mice, but not always correlated with increased P-PDH levels (Kiilerich et al., AJP-RICP, 2010).

Reviewer rebuttal: I am still not convinced that glycolysis is activated by PGC1alpha: in Kiilerich et al., the reduction in post-exercise muscle glycogen is the smallest in transgenic mice, both relatively as well as absolutely. The VO₂ measurements in Calvo et al. indicate a strong preference for fatty acid oxidation in the same mice. Summermatter et al. PNAS 2013 observed a strong reduction in anaerobic glycolysis in these animals. Finally, according to the Randle Cycle, the potent effect of PGC1alpha on lipid oxidation (which nobody disputes, at least to my knowledge) should throttle glucose oxidation. Thus, glycolytic flux would have to be measured in muscle to make a convincing case.

Original Point 3.) It is somewhat surprising that Kyn administration to control mice fails to have an effect (e.g. see Fig. 2 and S6), even though endogenous PGC1alpha expression in skeletal muscle is considerable even in the absence of transgenic overexpression. How do the authors explain this observation? It might be interesting to exercise Kyn-treated control mice, which then could be expected to exhibit an effect? Along the same lines, absence of a positive effect of Kyn treatment in PGC1alpha knockouts is expected if PGC1alpha were necessary for this. The authors however report a negative effect. What is the hypothesis for this?

Authors reply: Since muscle Kat expression is dependent on PGC-1a1 (Agudelo et al., Cell 2014), MKO and wt myotubes will have a reduced capacity to convert Kyn into Kyna, thus allowing its degradation through the kynurenine pathway. Interestingly, KP metabolites such as 3-hydroxykynurenine and 3-hydroxyanthranilic acid have been shown to impair energy metabolism by direct inhibition of mitochondrial respiration and ROS production Nagamure et al. Adv. Exp. Med. Biol. 1999; Okuda et al. J. Neurochem. 1998).

Reviewer rebuttal: I am not sure how this reply addresses my point: wt myotubes do express endogenous PGC1alpha, as does wt skeletal muscle in vivo. Shouldn't Kyn then have at least some effect? If not, what would be the physiological relevance of the reported observations if transgenic PGC1alpha were required?

Original Point 4.) It is surprising that in many of the experiments, gain- and loss-of-function of PGC1alpha fails to significantly affect the expression of many genes that have previously been reported to be altered in the same in vitro and in vivo experimental systems, for example, most of the transcription factors in Fig. 2 and S5, and many of the mitochondrial genes shown in Fig. S5

and S6 (shown in numerous papers). In fact, some of these genes appear to be significantly elevated in the transcriptomics analysis (Supplemental Table 1)?

Authors reply: We thank the reviewer for his/her point. We wonder if the y-axis might be misleading so if needed, we can replot the graphs. We can observe an increase in Esrra, PPAR α , NRF1, NRF2 almost 2-fold and 1.5-fold for MEF2 in both Mck-PGC-1 α 1 and myotubes transduced with PGC-1 α 1 adenovirus in accordance with what was described before (e.g. Lin et al. Nature 2002; Ruas et al. Cell 2012; Agudelo et al. Cell 2014). In a similar way, we also observe a decrease in the expression of several transcription factors in MKO-PGC-1 α mice (0.5- to 0.8-fold), similar to what was reported before (Handschin et al. J Biol Chem 2007; Handschin et al. J Clin Invest 2007).

Reviewer rebuttal: The y-axis was not misleading, but the absence of any statistical significance in the original figure. Are these the same data as reported in the first version of the paper? What kind of statistics was used to assess significance? Anova? If so, the significant differences of genotype and treatment should be indicated with different symbols.

Original Point 5.) To judge the results of the training study, the authors should indicate the time of sacrifice of the mice after the last bout of running wheel activity to discern acute from chronic effects. Moreover, some measure of the running activity of these mice would be helpful to assess the training.

Authors reply: We thank the reviewer for raising this important point. We have clarified in our methods the time of sacrifice of the mice (12 h after the last bout of exercise). In addition, we enclose the running activity of the mice when performing voluntary exercise. We can add this information to the manuscript if necessary.

Reviewer rebuttal: I do think that adding this information is important: sacrifice 12 hours after exercise would most likely still include acute effects of the training bout, which is important for the interpretation of the findings.

Reviewer #4:

Remarks to the Author:

I am happy to read the authors of the paper have taken most of my considerations into account and I feel the paper in its present form is a much improved version of the original submission.

Reviewer #5:

Remarks to the Author:

In the first round of review, the first comment was 'However, there is no evidence presented in the current manuscript that PGC-1 α is required for the effects of exercise. This needs to be addressed and could be done so through using PGC-1 α knockout mice.' This pertained to MAS responses following exercise in the PGC1 knockout. The authors responded with references showing the muscle fibre type, mitochondrial and exercise capacity adaptations in this model (and the transgenic) using prior literature. Perhaps there was a point of confusion, but the comment was focused on the importance of showing that the PGC1 α knockout models have altered MAS responses to exercise. Figures 5 and S9 do not show these responses in the knockout. This would be critical data to support the current conclusion as stated: 'Our findings show that PGC-1 α 1 activates the MAS in skeletal muscle, supported by kynurenine catabolism, as part of the adaptations to endurance exercise'.

Regarding the 2nd comment in the first round of review, the authors have clarified to the reviewer that the transgenic model have previously been shown to increase PGC α expression up to 10 fold higher depending on the muscle. This response to the reviewer should be incorporated into the manuscript as a discussion point noting the limitations of translating such large increases to a physiologically relevant context.

All other comments have been addressed and the paper clarified accordingly.

We would like to thank the reviewers for their time and constructive comments and suggestions to our previous revision. Please find below a point-by-point response:

Reviewers' comments:

Reviewer #1 (Remarks to the Author):

In the present version of the paper by Agudelo et al. the critical points suggested to be clarified have been rigorously addressed. This work shows in an extensive way that PGC-1 α 1 activates the MAS in skeletal muscle, supported by Kyn utilization, as part of the adaptations to endurance exercise. In my opinion the paper is well executed, solidly argued and has enough relevance and novelty for its publication. However, it is rather intriguing that silencing Slc25a12 in myotubes induces a 3-fold increment in aspartate content (figure S7i) whereas silencing Got2/KAT4 induces a clear reduction in aspartate (Figure 3i). It has been previously reported by different groups that there is a dramatic reduction of aspartate in mouse C2C12 myoblast AGC1/ARALAR knocked-down (in which AGC1 levels were about 8-fold higher than AGC2/CITRIN levels; Alkan HF et al., Cell Metab 2018), in brain and neurons from AGC1/ARALAR knockout mice (Jalil MA et al., JBC 2005) and in human brain (Wibom R et al., N. Engl J Med 2009).

We have included the reviewer's comments as a discussion point in the text (page 5, highlighted). In there we cite the references the reviewer indicates, and discuss that the different cell systems used have very different PGC-1a levels, and therefore probably Kat enzyme levels, which will undoubtedly affect the total outcome of this system. For example, the Alkan HF et al Cell Metab 2018 paper uses undifferentiated C2C12 myoblasts (which have low PGC-1a levels), under 10% FBS whereas we use fully differentiated myotubes that are grown under 2%HS. That brings the additional factor that the amounts of kynurenine in the system are most likely different as well.

Another question to be further discussed is that aspartate levels herein measured by using Abcam comercial kit (Figure S7i) are much higher than those reported in mice neurons and brain (Jalil MA et al., JBC 2005). Is there any reference available about aspartate levels measured with other methods (amino acid analyzer or HPLC/GC) in primary myotubes or mice skeletal muscle?

The reviewers are correct. The units in graph should be nmol/mg protein. We apologize for mistake and this is now corrected. The (corrected) values are in line with what has previously been reported for human muscle

(Essén-Gustavsson E. & Blomstrand E. Acta Physiol Scand 2002) and this information has also been added to the text (page 5, highlighted).

Minor verification: In Figure 1K, it is represented Got2/Kat3 instead of Got2/Kat4.

We thank the reviewers for pointing this out. We have corrected this in the figures and throughout the manuscript.

Reviewer #2 (Remarks to the Author):

Original Point 1.) The authors should emphasize the conceptual advances of their study. First, due to the strong effect of PGC1alpha on mitochondrial biogenesis and function, and increase in gene expression, most of which are mitochondrial, might be expected. While I readily acknowledge that “expected” is not show, second, more importantly, the control of the malate aspartate shuttle as well as aspartate and glutamate levels has previously been reported (Hatazawa et al. PLOS ONE 2015, PMID 26114427 – a study which probably should be cited here).

Authors reply: Indeed, one of the data sets that we used for our initial multi-omics analysis was the one reported by Hatazawa et al., in PLOS One 2015. This is stated right at the beginning of the manuscript and the article is cited there and then again in methods under “transcriptomic and metabolomic analysis”. We have now also mentioned it in a new paragraph of concluding remarks. Our message is not only that PGC-1a1 activates the MAS in muscle, showing here the gene networks involved, and data from in vitro and in vivo manipulation of the system. It is also that in the trained state (i.e. high PGC-1a1 levels), skeletal muscle can use Kyn (a compound with known neurotoxic properties) to support energy metabolism. We also show that in the absence or at low levels of PGC-1a, Kyn (or its metabolites) has mostly toxic effects, even inhibiting mitochondrial respiration. We have highlighted this information throughout the manuscript.

Reviewer rebuttal: There might have been a misunderstanding here: my intention was not to criticize an absence of citation of Hazazawa et al., but a lack of attribution of their findings in regard to PGC1alpha regulation of the malate and aspartate shuttle as well as aspartate and glutamate levels, which also is a central aspect of the current manuscript. The authors have now partially rectified this by adding the sentence “The same was observed in the muscle metabolomics data reported for the HSA-PGC-1 α -b mice⁵”, referring to the elevation of glutamate and aspartate. I fail to see reference 5 in the new

paragraph of concluding remarks as stated by the authors in their reply. I agree that the other points regarding novelty are now more clearly highlighted in the text.

We apologize for the mistake. This has been corrected, and additional clarification and the citation have been added to the discussion (page 7, highlighted).

Original Point 2.) The authors link their postulated mechanism to an increase in glycolysis. However, most prior data revealed that PGC1alpha strongly reduces glycolysis in muscle, e.g. as demonstrated by Wende et al. JBC 2007, PMID 17932032, who have demonstrated reduced glycolytic flux and expression of phosphofructokinase in PGC1alpha inducible muscle-specific transgenic mice. Downstream, PGC1alpha-controlled elevation of PDK4 would further throttle glucose oxidation (Wende et al. Mol Cell Biol 2005, PMID 16314495). Thus collectively, the PGC1alpha-mediated increase in glucose uptake leads to a rerouting of glucose to glycogen and the pentose phosphate pathway, but not glycolysis. How do the authors reconcile their hypothesis with these observations?

Authors reply: The reviewer is correct in pointing out that different mouse models of elevated PGC-1a1 expression in skeletal muscle have not always generated overlapping results (Mck-PGC-1a1, Lin J. et al. Nature 2002; PGC-1a TRE (+), Wende et al. JBC 2007; HSA-PGC-1a-b, Miura et al., Endocrinology 2008). The 3 models use different strategies to elevate PGC-1a in muscle. Mck- and HSA-driven PGC-1a expression is already activated at birth and remains on during the life of the mice, whereas the model referred to by the reviewer is an inducible model (mice are kept on doxycycline, which represses transgene expression until it is removed from the diet). Evidence of differences between the models is reflected on exercise performance, Glut4 levels, glucose tolerance, insulin sensitivity, among other. Namely, the mouse model that the reviewer refers to (PGC-1a TRE (+)) has been shown to have a reduction in exercise capacity (Wende et al., JBC 2017), whereas the Mck-PGC-1a mouse has increased exercise performance (Calvo et al., JAP 2008). Interestingly, both models show increased muscle glycogen content at rest but the Mck-PGC-1a can use it during exercise (Kiilerich et al., AJP-RICP, 2010) whereas the PGC-1a TRE (+) cannot (Wende et al., JBC 2017). As it is very difficult to compare the levels of PGC-1a overexpression between the papers/models, we can only say that they fundamentally differ on chronic vs acute PGC-1a elevation. In addition, the effect of PGC-1a on PDK4 referred to by the reviewer, which was observed in C2C12 cells, was also observed in the Mck-PGC-1a1 mice, but not always correlated with increased P-PDH levels (Kiilerich et al., AJP-

RICP, 2010).

Reviewer rebuttal: I am still not convinced that glycolysis is activated by PGC1alpha: in Kiilerich et al., the reduction in post-exercise muscle glycogen is the smallest in transgenic mice, both relatively as well as absolutely. The VO₂ measurements in Calvo et al. indicate a strong preference for fatty acid oxidation in the same mice. Summermatter et al. PNAS 2013 observed a strong reduction in anaerobic glycolysis in these animals. Finally, according to the Randle Cycle, the potent effect of PGC1alpha on lipid oxidation (which nobody disputes, at least to my knowledge) should throttle glucose oxidation. Thus, glycolytic flux would have to be measured in muscle to make a convincing case.

We have now measured glycolytic flux using seahorse respirometry in myotubes with or without PGC-1 α 1 expression (Fig S4). These results show that ECAR values (indicative of lactate secretion to the media) are higher in myotubes expressing PGC-1 α 1. Also, in PGC-1 α 1 myotubes, oligomycin further elevated ECAR, compared to the control cells.

Original Point 3.) It is somewhat surprising that Kyn administration to control mice fails to have an effect (e.g. see Fig. 2 and S6), even though endogenous PGC1alpha expression in skeletal muscle is considerable even in the absence of transgenic overexpression. How do the authors explain this observation? It might be interesting to exercise Kyn-treated control mice, which then could be expected to exhibit an effect? Along the same lines, absence of a positive effect of Kyn treatment in PGC1alpha knockouts is expected if PGC1alpha were necessary for this. The authors however report a negative effect. What is the hypothesis for this?

Authors reply: Since muscle Kat expression is dependent on PGC-1 α 1 (Agudelo et al., Cell 2014), MKO and wt myotubes will have a reduced capacity to convert Kyn into Kyna, thus allowing its degradation through the kynurenine pathway. Interestingly, KP metabolites such as 3-hydroxykynurenine and 3-hydroxyanthranilic acid have been shown to impair energy metabolism by direct inhibition of mitochondrial respiration and ROS production Nagamure et al. Adv. Exp. Med. Biol. 1999; Okuda et al. J. Neurochem. 1998).

Reviewer rebuttal: I am not sure how this reply addresses my point: wt myotubes do express endogenous PGC1alpha, as does wt skeletal muscle in vivo. Shouldn't Kyn then have at least some effect? If not, what would be the physiological relevance of the reported observations if transgenic PGC1alpha were required?

The mck-PGC-1 α transgenic elevates overall muscle PGC-1 α levels to those observed in wild-type soleus (Lin J et al, Nature 2002). So even in the TG mice, we are working with PGC-1 α levels that can be seen in a strongly oxidative fiber. We chose the Gastrocnemius muscle for gene expression and protein determination, since it's a mixed fiber muscle and where there is still possibility to increase PGC-1 α levels with, for example, exercise training (or transgenic expression). If one looks at the PGC-1 α levels (and related changes) in the mck-PGC-1 α soleus there are very few changes (since in that muscle, TG expression of PGC-1 α doesn't elevate it much above endogenous levels).

We think that, for that reason, in WT myotubes (derived from myoblasts), where the endogenous PGC-1 α levels are low, and in white muscles that also have lower levels of PGC-1 α , the effects of kynurenine are not obvious since we are using low kyn doses (within physiological levels) and the expression of the KAT / MAS system is low.

This has been added to the discussion (page 4, highlighted)

Original Point 4.) It is surprising that in many of the experiments, gain- and loss-of-function of PGC1alpha fails to significantly affect the expression of many genes that have previously been reported to be altered in the same in vitro and in vivo experimental systems, for example, most of the transcription factors in Fig. 2 and S5, and many of the mitochondrial genes shown in Fig. S5 and S6 (shown in numerous papers). In fact, some of these genes appear to be significantly elevated in the transcriptomics analysis (Supplemental Table 1)?

Authors reply: We thank the reviewer for his/her point. We wonder if the y-axis might be misleading so if needed, we can replot the graphs. We can observe an increase in Esrra, PPARa NRF1, NRF2 almost 2-fold and 1.5-fold for MEF2 in both Mck-PGC-1a1 and myotubes transduced with PGC-1a1 adenovirus in accordance with what was described before (e.g. Lin et al. Nature 2002; Ruas et al. Cell 2012; Agudelo et al. Cell 2014). In a similar way, we also observe a decrease in the expression of several transcription factors in MKO-PGC-1a mice (0.5- to 0.8-fold), similar to what was reported before (Handschin et al. J Biol Chem 2007; Handschin et al. J Clin Invest 2007).

Reviewer rebuttal: The y-axis was not misleading, but the absence of any statistical significance in the original figure. Are these the same data as reported in the first version of the paper? What kind of statistics was used to assess significance? Anova? If so, the significant differences of genotype and treatment should be indicated

with different symbols.

The absence of indication of statistical significance in the original figure was a lapse on our side. It wasn't that there wasn't statistical significance, we forgot to add that indication to the figure. We apologize for that omission which has now been corrected. The statistics used to assess the significance is indicated in the figure legend (page 9, highlighted).

Original Point 5.) To judge the results of the training study, the authors should indicate the time of sacrifice of the mice after the last bout of running wheel activity to discern acute from chronic effects. Moreover, some measure of the running activity of these mice would be helpful to assess the training.

Authors reply: We thank the reviewer for raising this important point. We have clarified in our methods the time of sacrifice of the mice (12 h after the last bout of exercise). In addition, we enclose the running activity of the mice when performing voluntary exercise. We can add this information to the manuscript if necessary.

Reviewer rebuttal: I do think that adding this information is important: sacrifice 12 hours after exercise would most likely still include acute effects of the training bout, which is important for the interpretation of the findings.

This is clarified in the methods and we have now also included it in the corresponding Figure legend (page 10, highlighted).

Reviewer #4 (Remarks to the Author):

I am happy to read the authors of the paper have taken most of my considerations into account and I feel the paper in its present form is a much improved version of the original submission.

We are happy that the reviewer is satisfied with the revised version of the manuscript.

Reviewer #5 (Remarks to the Author):

In the first round of review, the first comment was 'However, there is no evidence presented in the current manuscript that PGC-1 alpha is required for the effects of exercise. This needs to be addressed and could be done so through using PGC-1 alpha knockout mice.' This pertained to MAS responses following exercise in the PGC1 knockout. The authors responded with references showing the muscle fibre type, mitochondrial and exercise capacity adaptations in this model (and the transgenic) using prior literature. Perhaps there was a point of confusion, but the comment was focused on the importance of

showing that the PGC1alpha knockout models have altered MAS responses to exercise. Figures 5 and S9 do not show these responses in the knockout. This would be critical data to support the current conclusion as stated: 'Our findings show that PGC-1 α 1 activates the MAS in skeletal muscle, supported by kynurenine catabolism, as part of the adaptations to endurance exercise'.

We have now included new data showing that muscle-specific PGC-1a KO mice (MKO-PGC-1a) do not upregulate MAS genes upon acute or chronic exercise (Figure 5d and S9c). We thank the reviewer for the suggestion and clarification which indeed has further supported our conclusions.

Regarding the 2nd comment in the first round of review, the authors have clarified to the reviewer that the transgenic model have previously been shown to increase PGCalpha expression up to 10 fold higher depending on the muscle. This response to the reviewer should be incorporated into the manuscript as a discussion point noting the limitations of translating such large increases to a physiologically relevant context.

This information is included in page 2 (highlighted) and to put it into physiological context we cite Mathai et al (JAP 2008) who report human data showing an increase of 7 to 9-fold in PGC-1a levels in the vastus lateralis following a cycling to exhaustion.

All other comments have been addressed and the paper clarified accordingly.

Reviewers' Comments:

Reviewer #1:

Remarks to the Author:

The authors have satisfactorily addressed my suggestions, and I believe that the current version is clearly improved for publication.

I just want to add a clarification for the authors about point 1. I assumed that silencing Slc25a12 expression (with functional Kat1) induces an accumulation of cytosolic glutamate and reversion of Got1 enzyme to generate aspartate in the cytosol and that is why aspartate is increased after silencing Slc25a12 by 44% (Figure S7i). Whereas, in the same experimental system with myotubes, kat inhibition with carbidopa (Figure 4a-c) or silencing Got2/Kat4 expression (figure 3h and 3i) induced the "expected" reduction in OCR and aspartate levels.

Reviewer #5:

None

REVIEWERS' COMMENTS:

Reviewer #1 (Remarks to the Author):

The authors have satisfactorily addressed my suggestions, and I believe that the current version is clearly improved for publication.

I just want to add a clarification for the authors about point 1. I assumed that silencing Slc25a12 expression (with functional Kat1) induces an accumulation of cytosolic glutamate and reversion of Got1 enzyme to generate aspartate in the cytosol and that is why aspartate is increased after silencing Slc25a12 by 44% (Figure S7i). Whereas, in the same experimental system with myotubes, kat inhibition with carbidopa (Figure 4a-c) or silencing Got2/Kat4 expression (figure 3h and 3i) induced the “expected” reduction in OCR and aspartate levels.

We have added this to Supplementary Fig. 9a